# Safe eradication of large established tumors using neovasculature-targeted tumor necrosis factor-based therapies

Leander Huyghe[1], Alexander Van Parys[1,¶], Anje Cauwels[1,¶], Sandra Van Lint[1], Stijn De Munter[2], Jennyfer Bultinck[1,†], Lennart Zabeau[3], Jeroen Hostens[4,‡], An Goethals[4,§], Nele Vanderroost[1], Annick Verhee[1], Gilles Uzé[5], Niko Kley[3], Frank Peelman[6], Bart Vandekerckhove[2], Peter Brouckaert[4] & Jan Tavernier[1,3,*] [ID]

## Abstract

Systemic toxicities have severely limited the clinical application of tumor necrosis factor (TNF) as an anticancer agent. Activity-on-Target cytokines (AcTakines) are a novel class of immunocytokines with improved therapeutic index. A TNF-based AcTakine targeted to CD13 enables selective activation of the tumor neovasculature without any detectable toxicity *in vivo*. Upregulation of adhesion markers supports enhanced T-cell infiltration leading to control or elimination of solid tumors by, respectively, CAR T cells or a combination therapy with CD8-targeted type I interferon AcTakine. Co-treatment with a CD13-targeted type II interferon AcTakine leads to very rapid destruction of the tumor neovasculature and complete regression of large, established tumors. As no tumor markers are needed, safe and efficacious elimination of a broad range of tumor types becomes feasible.

**Keywords** cancer; interferons; neovasculature; targeted therapy; tumor necrosis factor

**Subject Categories** Cancer; Immunology; Pharmacology & Drug Discovery

See also: **T Kammertoens et al** (February 2020)

## Introduction

Tumor necrosis factor (TNF) can cause rapid hemorrhagic tumor necrosis in both animal models and patients. Its therapeutic use, however, is severely impeded by life-threatening side-effects, i.e., systemic inflammatory response syndrome that may lead to shock and organ failure. In clinical studies, doses up to 50 times below the estimated effective dose already caused serious side-effects, and doses close to the maximum tolerated dose produced only minimal tumor responses (Roberts *et al*, 2011). The current clinical use of TNF is therefore limited to the treatment of advanced cancers in the limbs via isolated limb perfusion (ILP). The high complete response rates achieved by ILP with TNF (2–6 mg) in combination with melphalan clearly show the potential of TNF as an anticancer drug (Eggermont *et al*, 2003; Lejeune *et al*, 2006; van Veenendaal *et al*, 2017). The exact mechanism underlying the antitumor effect of TNF is still poorly understood but several lines of evidence point to a critical role of the TNF-R1 receptor complex on tumor neovasculature (Van de Wiel *et al*, 1989; Ruegg *et al*, 1998; Stoelcker *et al*, 2000; Lejeune *et al*, 2006).

One strategy to reduce cytokine systemic toxicity is fusions to targeting moieties (e.g., antibody fragments, single-domain antibodies or peptides) that direct cytokine activity to a specific cell or tissue, so that their therapeutic index is increased by lowering the effective dose (List & Neri, 2013). Following this strategy, TNF fusion proteins NGR-TNF, targeting the neovasculature marker aminopeptidase N/CD13, and L19-TNF, targeting an oncofetal splice variant of fibronectin, are currently in clinical development for the treatment of mesothelioma and melanoma, respectively (Danielli *et al*, 2015a; Gregorc *et al*, 2018). However, the high affinity receptor binding,

1 Cytokine Receptor Laboratory, VIB Center for Medical Biotechnology, Department of Biomolecular Medicine, Ghent University, Ghent, Belgium
2 Department of Clinical Chemistry, Microbiology and Immunology, Ghent University, Ghent, Belgium
3 Orionis Biosciences, Boston, MA, USA
4 Department of Biomedical Molecular Biology, Ghent University, Ghent, Belgium
5 CNRS UMR 5235, University of Montpellier, Montpellier, France
6 VIB Center for Medical Biotechnology, Department of Biomolecular Medicine, Ghent University, Ghent, Belgium
*Corresponding author. Tel: +32 9 264 9302; Fax: +32 9 264 9340; E-mail: jan.tavernier@vib-ugent.be
†Present address: Oxyrane, Ghent, Belgium
‡Present address: PerkinElmer, Zaventem, Belgium
§Present address: Genae, Antwerp, Belgium
¶Shared authorship

which is intrinsic to wild-type (wt) cytokines, inevitably entails the high risk of side-effects, especially in the case of pleiotropic cytokines. On top, a large portion of the wt cytokine fusion drugs will be captured by their ubiquitously expressed receptors before they can reach their target cells (the so-called "sink effect" (Tzeng *et al*, 2015)) and, as is the case for TNF, also by soluble receptors in circulation. To avoid such unwanted effects, we have engineered Activity-on-Target cytokines (AcTakines), in which the wt cytokine is replaced by a mutant version with strongly reduced affinity for its receptor, and fused to a targeting moiety that binds a cell-specific surface marker. Consequently, AcTakines are inactive on their way to the target, thereby eliminating the unfavorable side and sink effects, but regain full activity on the targeted cells by local avidity-driven receptor binding (Garcin *et al*, 2014). We have previously reported the remarkable efficacy of a dendritic cell-targeted type I interferon AcTakine (AcTaferon, AFN) in multiple tumor models, without side-effects (Cauwels *et al*, 2018a,b). In this study, we now demonstrate the efficacy of tumor vasculature-targeted TNF and IFN-γ AcTakines (resp. AcTafactor, AFR; and AcTaferon-II, AFN-II) to selectively activate or kill tumor endothelial cells. Tumor vasculature-targeted AFR therapy synergized strikingly with either mouse CD8-AFN immunotherapy, human CAR T-cell immunotherapy, or tumor vasculature-targeted AFN-II therapy, leading to complete eradication of large established tumors in mice without toxicity.

# Results

## TNF activity on endothelial cells is safe and sufficient to induce tumor necrosis

To investigate the importance of TNF-R1 on endothelial cells for the antitumor activity of TNF, we made use of mice having a conditional reactivation mutant allele for TNF-R1 (p55$^{\text{cneo/cneo}}$ mice) in which TNF-R1 expression is restored upon Cre-mediated excision of an inhibitory floxed Neo cassette (Victoratos *et al*, 2006). Crossing p55$^{\text{cneo/cneo}}$ mice with Flk1Cre mice yielded mice that express functional levels of TNF-R1 on endothelial cells only (Fig 1C). Daily perilesional (p.l.) treatment of wt C57BL/6 mice bearing B16Bl6 melanoma tumors with 7 µg mTNF resulted within 2–3 days in extensive tumor necrosis (Fig 1B). A similar effect was observed in deleter Cre (DelCre) p55$^{\text{lox/lox}}$ control mice. Despite having a TNF-R1-expressing tumor, the antitumor effect of TNF was completely absent in either TNF-R1$^{-/-}$ or p55$^{\text{cneo/cneo}}$ mice. Conversely, when wt mice carrying a TNF-unresponsive B16Bl6 tumor (B16-dnTNF-R1) were treated with TNF, the antitumor effect was similar to that observed with a parental B16Bl6 tumor (Fig 1A). Together these data unequivocally demonstrate that the antitumor effect of TNF is host-mediated. Significantly, when Flk1Cre p55$^{\text{cneo/cneo}}$ mice were treated with TNF, the antitumor effect was identical to that induced in wt mice, indicating that TNF signaling through TNF-R1 on endothelial cells is sufficient to induce its antitumor effect.

To investigate the safety of endothelial TNF activity, we used a bolus shock model. Intravenous (i.v.) injection of a lethal dose of TNF causes a drastic drop in body temperature, systemic inflammation, shock, and eventually death within 12–72 h. In naive wt C57BL/6 mice, the LD50 and LD100 of mTNF were 6 and 10 µg, respectively (Fig 1D). As expected, p55$^{\text{cneo/cneo}}$ mice were

completely resistant to TNF toxicity. The sensitivity to TNF was slightly reduced in DelCre p55$^{\text{lox/lox}}$ mice, with an LD50 and LD100 of 10 and 25 µg, respectively, reflecting either incomplete Cre-mediated reactivation of the conditional TNF-R1 allele or differences in genetic background. In stark contrast, however, Flk1Cre p55$^{\text{cneo/cneo}}$ mice were completely resistant to TNF-induced shock, with doses up to 250 µg not even causing a drop in body temperature. Together these data show that focusing TNF activity selectively to endothelial cells is potentially safe and effective as tumor therapy.

## AcTafactors allow target-specific delivery of TNF activity

An AcTakine typically consists of three parts: a mutant cytokine with strongly reduced binding affinity for its receptor complex, a flexible linker, and a VHH-type targeting moiety (Fig 2A). Given the homotrimeric nature of TNF, we opted to use a single-chain version of TNF (scTNF), with short C- to N-terminal linkers between the TNF monomers (Krippner-Heidenreich *et al*, 2008). This genetic linkage precludes *in vivo* monomer exchange with wtTNF. To generate AcTafactors (AFRs), we evaluated an extensive panel of mouse scTNF mutants fused to a VHH directed against mouse CD20. The Y86F mutation reduced the biological activity in an L929 cytotoxicity assay by about 10 000-fold (Fig 2B). Strikingly, the activity of this mCD20-AFR was largely recovered when assayed on L929 cells stably expressing mCD20. Such AcTakine effect was absent using a non-targeted VHH control BcII10-AFR. Similarly, we generated a human AFR by fusing the human scTNF Y87F mutant to a VHH directed against hCD20. When assayed on MCF7 cells, this hCD20-AFR displayed a 280-fold reduction in cytotoxic activity, while on MCF7-hCD20 cells a nearly complete recovery of biological activity was observed (Fig 2C). In a more physiological setting, we next used primary human umbilical vein endothelial cells (HUVECs) and hCD13, which is naturally expressed on these cells, as target. As HUVECs are not susceptible to TNF-induced cell death, we used NF-κB-dependent IL-8 secretion as readout. At concentrations up to 20 ng/ml, the non-targeted human scTNF mutant Y87F had virtually no effect on HUVECs, while the hCD13-targeted AFR induced significant IL-8 secretion (Fig 2D). Interestingly, even at high concentration hCD13-AFR was unable to induce the IL-8 secretion level of wtTNF, suggesting that the signaling cascade initiated by AFR might differ from that by wtTNF (see also below).

## AFRs combine safety and activity *in vivo*

To evaluate *in vivo* toxicity, we first used the i.v. shock model in naive C57BL/6 mice. The BcII10-AFR was used to avoid target-specific effects. It was previously reported by Krippner-Heidenreich *et al* (2008) that scTNF displayed decreased toxicity compared to trimeric wtTNF. Indeed, injection of 10 µg scTNF, which is lethal when using wtTNF, only led to a moderate drop in body temperature (Fig 3A), but induced significant levels of circulating IL-6, a sensitive marker for systemic TNF activity (Fig 3B). Thirty-five microgram (1.75 mg/kg) of scTNF was an LD100 in this model and was characterized by a dramatic drop in body temperature and concomitant increase in circulating IL-6. For BcII10-AFR, 5 consecutive daily i.v. bolus injections of 200 µg (10 mg/kg) were completely non-toxic and did not induce any drop in body temperature or increase in circulating IL-6, confirming that untargeted AFRs are inactive.

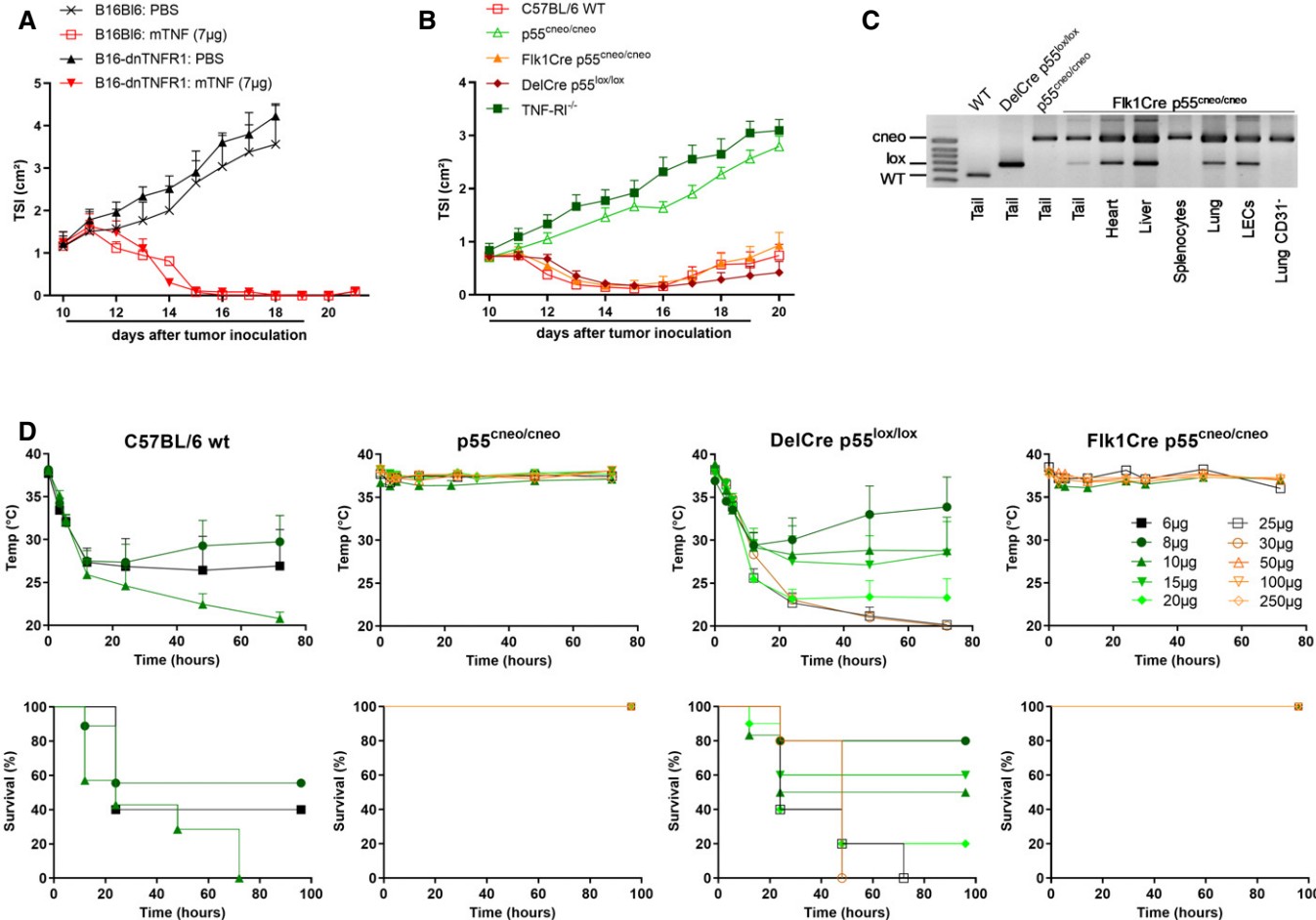

**Figure 1. TNF-R1 on tumor vasculature is sufficient for the antitumor effect of TNF, but not for its shock-inducing effect.**

A  C57BL/6 mice were inoculated with $6 \times 10^5$ B16Bl6 parental or dnTNF-R1 cells on day 0 and treated daily with 7 μg mTNF p.l. from day 10. Tumor growth is shown as mean tumor size index (TSI) + SEM ($n = 6$). The line under the graph represents the treatment period.

B  B16Bl6 tumor growth after daily p.l. treatment with 7 μg mTNF in C57BL/6J wild-type (WT), TNF-R1$^{-/-}$, conditional TNF-R1 reactivation knockout (p55$^{cneo/cneo}$), or p55$^{cneo/cneo}$ mice with Cre expression in endothelium (Flk1Cre) or all cells (DelCre). Tumor growth is shown as mean TSI + SEM ($n = 7$ for WT and Flk1Cre, 8 for DelCre, 9 for TNF-R1$^{-/-}$, and 10 for p55$^{cneo/cneo}$). The line under the graph represents the treatment period.

C  PCR analysis for the detection of wild-type, conditional knockout (cneo) or reactivated (lox) TNF-R1 allele in whole tissues, lung endothelial cells (LECs), and lung single-cell suspension depleted of CD31$^+$ cells.

D  Toxicity of a single i.v. bolus injection of the indicated dose of mTNF. Mean rectal body temperature + SEM and cumulative survival rates are shown ($n = 4$ for Flk1Cre 10 μg; 6 for DelCre 10 μg, Flk1Cre 10 μg, and p55$^{cneo/cneo}$ 15 μg; 7 for WT 10 μg; 9 for WT 8 μg; 10 for DelCre 20 μg; and 5 for all other groups). For continuity of the temperature graphs, dead mice were included with a temperature of 20°C.

Source data are available online for this figure.

To target tumor vasculature, we generated a VHH against mouse CD13 (Curnis *et al*, 2000; Pasqualini *et al*, 2000). This VHH was fused to wt scTNF or mutant Y86F. The immunocytokine mCD13-targeted wt scTNF was slightly more efficient in the B16Bl6 melanoma model than untargeted scTNF, confirming the targeting potential of this VHH. To demonstrate the *in vivo* activity of tumor vasculature-targeted AFRs, we performed immunohistochemical analysis of tumor sections after injection of wtTNF or mCD13-AFR (Fig 3C). Six hours after injection, clear induction of ICAM-1 on tumor vessels was observed, which was even more pronounced 24 h after injection, indicating prolonged endothelial activation. Induction of ICAM-1 expression was confirmed via qPCR analysis on whole tumor RNA (Fig 3D). Quite similar to wtTNF, mCD13-AFR induced expression of both ICAM-1 and E-selectin and repression of

VEGF-R2 at 4 h after injection. Since tumor-bearing mice tend to be sensitized toward TNF toxicity (Cauwels *et al*, 2018b), we next tested mCD13-AFR in the B16Bl6 melanoma model. In stark contrast to wt TNF, daily treatment for 10 consecutive days with a therapeutic dose of mCD13-AFR caused no significant weight loss or other apparent toxicity, confirming its safety (Fig 3E). The antitumor effect caused by mCD13-AFR was highly significant, while neither a non-targeted BcII10-AFR nor a fusion protein of mCD13 VHH and wt hIFNα2 (which does not bind the mouse IFNAR) had an effect. Interestingly, even at higher doses, mCD13-AFR appeared unable to induce tumor necrosis and vascular disruption to the same extent as wtTNF. We therefore investigated whether inhibition of pro-survival signaling could increase the efficacy of mCD13-AFR therapy. Pretreatment with either the Smac-mimetic Birinapant or the broad-

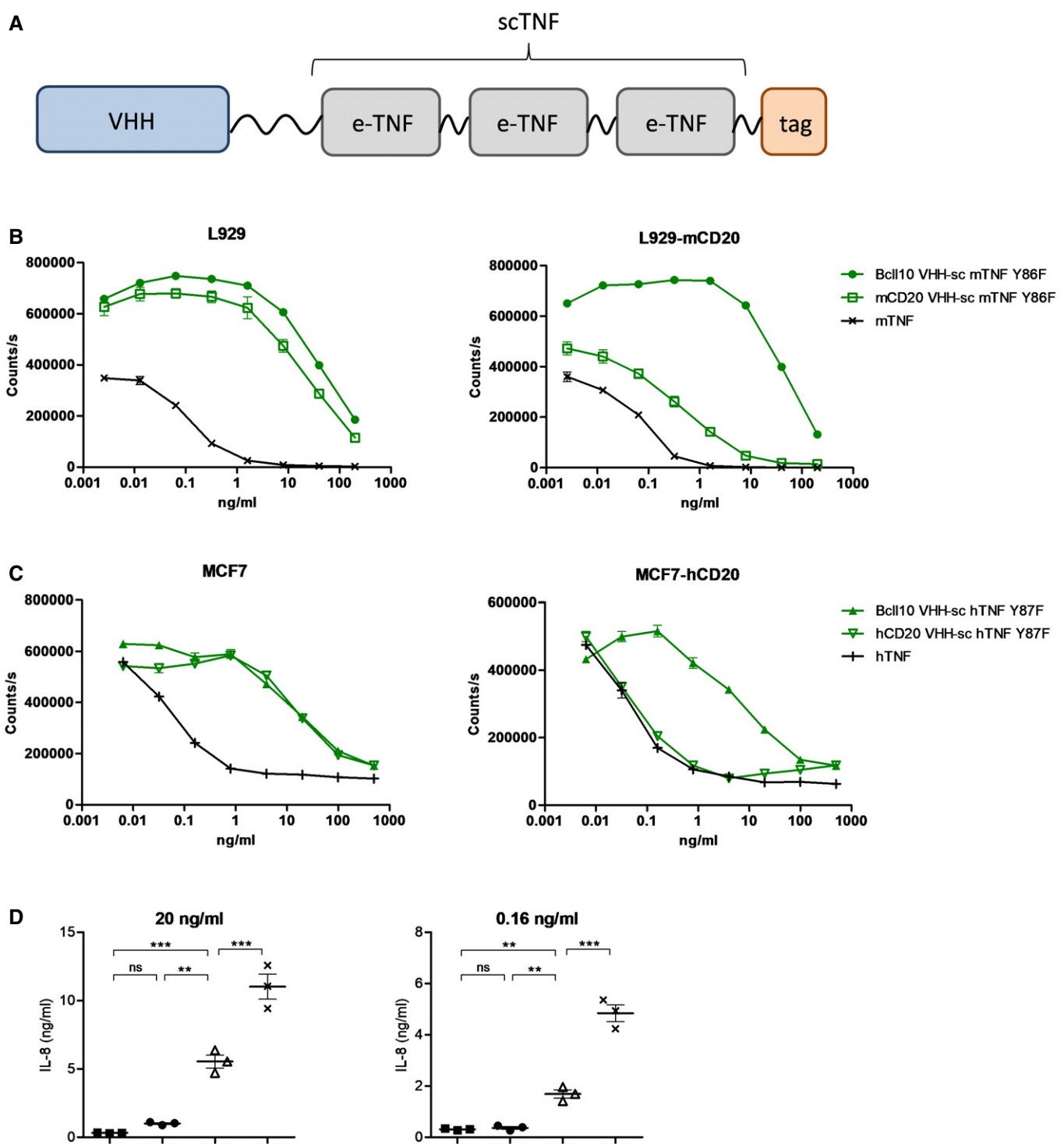

**Figure 2. AFRs mediate target-specific delivery of TNF activity.**

A    Schematic representation of AFR, consisting of N-terminal VHH fused via a 20xGGS linker to an engineered TNF (e-TNF) variant and C-terminal affinity tag.

B, C    Viability of L929 parental or mCD20-expressing cells after 72-h stimulation with mTNF or sc mTNF mutant Y86F fused to a BcII10 or mCD20 VHH (B) and viability of MCF7 parental or hCD20-expressing cells after 72-h stimulation with hTNF or sc hTNF mutant Y87F fused to a BcII10 or hCD20 VHH (C). Cell viability was measured via an ATP luminescence assay. Each point is the mean of three replicates, and error bars are SEM.

D    IL-8 concentration in HUVEC supernatant after 24-h stimulation with the indicated concentration of stimulus, measured via ELISA. Error bars are SEM. ns, non-significant; **P < 0.01; ***P < 0.001 by one-way ANOVA with Bonferroni's multiple comparison test.

Source data are available online for this figure.

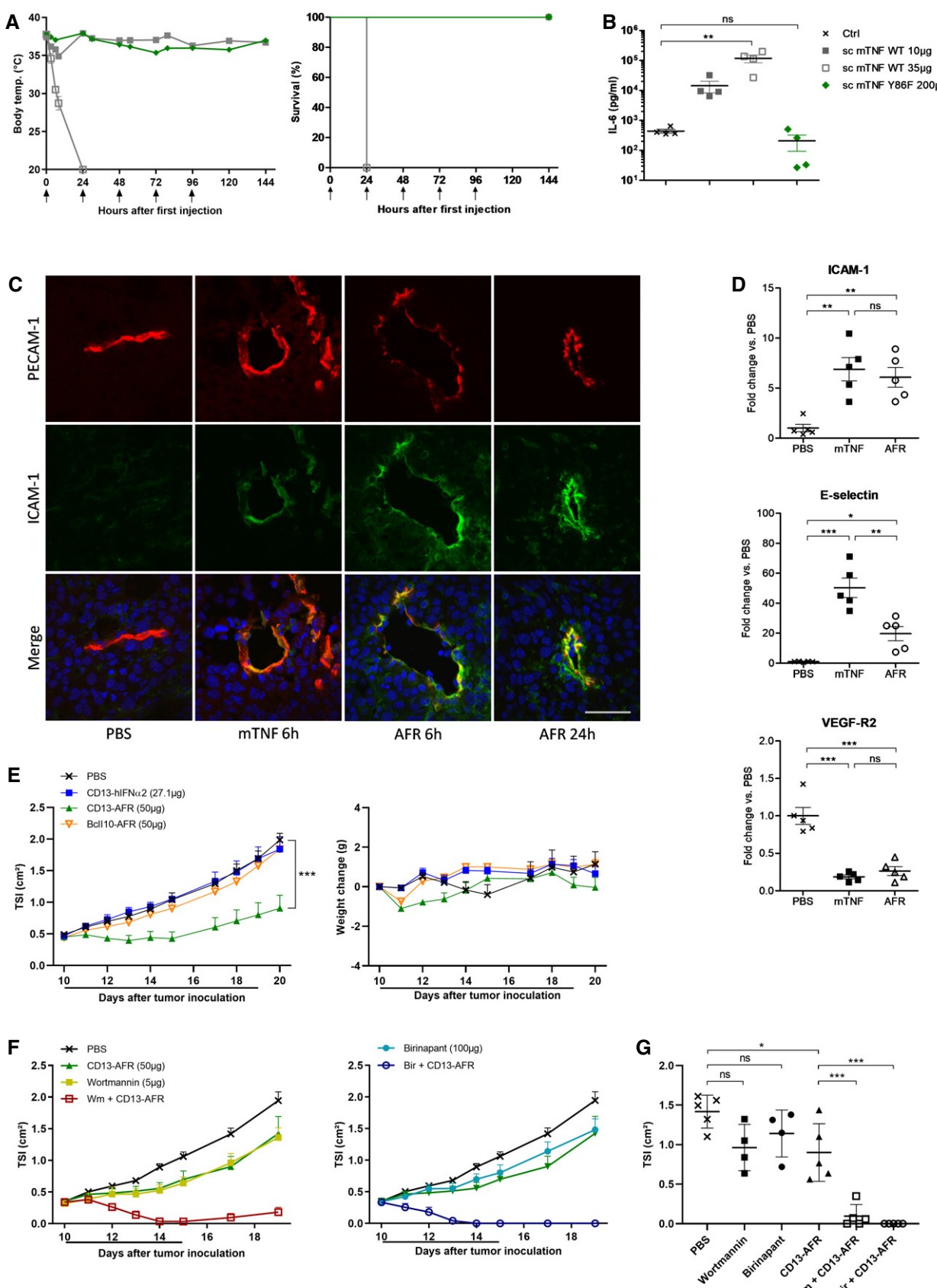

Figure 3.

**Figure 3. Sc mTNF Y86F is non-toxic and mCD13-AFR is active in mice.**

A  Toxicity of wt sc mTNF and mutant Y86F after injection (↑) in naïve C56BL/6 mice. First injection was i.v., and other injections were i.p. Mean rectal body temperature ± SEM and cumulative survival rates are shown (n = 4). For continuity of the temperature graphs, dead mice were included with a temperature of 20°C.

B  Plasma IL-6 concentrations measured via ELISA. Blood samples were taken 6 h after i.v. injection of wt sc mTNF or mutant Y86F. Values of individual mice and mean + SEM are shown. **$P < 0.01$; ns, non-significant by one-way ANOVA with Bonferroni's multiple comparison test.

C  Immunohistochemical staining for PECAM-1 (red), ICAM-1 (green), and DNA (blue) of B16Bl6 tumors after p.l. treatment with 7 μg wt mTNF or 50 μg of mCD13-AFR. Scale bar is 50 μm.

D  Expression of ICAM-1, E-selectin, and VEGF-R2 in B16Bl6 tumor 6 h after treatment, by qPCR analysis of whole tumor RNA. Error bars are SEM. ns, non-significant; *$P < 0.05$; **$P < 0.01$; ***$P < 0.001$ by one-way ANOVA with Bonferroni's multiple comparison test.

E  Tumor growth and body weight change after daily p.l. injection of the indicated (equimolar) doses of mCD13-AFR, non-targeted AFR (BcII10-AFR) or mCD13 VHH control (CD13-hIFNα2) in the B16Bl6 melanoma model. The line under the graph represents the treatment period. Tumor growth is shown as mean TSI, and error bars are SEM (n = 5). ***$P < 0.001$ by two-way ANOVA with Bonferroni's multiple comparison test.

F  B16Bl6 tumor growth after daily p.l. treatment with the indicated doses of mCD13-AFR, Wortmannin (Wm), Birinapant (Bir), or combinations thereof. Wortmannin and Birinapant were given 1 h before CD13-AFR. Tumor growth is shown as mean TSI + SEM (n = 4 for Wm and Bir, 5 for other groups). The line under the graph represents the treatment period.

G  TSI of individual tumors of the indicated treatment groups at day 17 after tumor inoculation. Error bars are SEM. ns, non-significant; *$P < 0.05$; *** $< 0.001$ by one-way ANOVA with Bonferroni's multiple comparison test.

Source data are available online for this figure.

spectrum PI3K inhibitor wortmannin clearly sensitized for the anti-tumor effect of mCD13-AFR, resulting in rapid and complete tumor necrosis, resembling the effect of wtTNF (Fig 3F and G). Taken together, these data suggest that mCD13-AFR therapy, in contrast to wtTNF, favored pro-inflammatory and survival signaling in tumor vasculature, which could be beneficial in combination treatments with chemo- or immunotherapy.

### Vasculature-targeted AFR synergizes with immunotherapy

We investigated whether AFR-induced activation of tumor vasculature would suffice to synergize with immunotherapies based on T-cell activation. CAR T-cell therapy has promising potential for the treatment of certain leukemias and lymphomas, but did not show clinical efficacy in the treatment of solid tumors. This is at least partly due to the poor tumor infiltration of CAR T cells (D'Aloia et al, 2018). Therefore, we evaluated whether mCD13-AFR-mediated activation of tumor vasculature could facilitate CAR T-cell infiltration and therapy in solid tumors. As a model, we used hCD70$^+$ SKOV3 human ovarian cancer cells that can be targeted using hCD70 CAR (Fig 4A) T cells in NSG mice. Treatment with mCD13-AFR, which on itself had no effect on SKOV3 tumor growth, clearly potentiated hCD70 CAR T-cell therapy, resulting in immediate tumor control by CAR T cells and tumor stasis (Fig 4B and C). By contrast, this combination therapy had no effect in the hCD70-negative RL model (Fig EV1D), underlining its antigen specificity. FACS analysis confirmed significantly increased CAR T-cell infiltration in SKOV3 tumors after AFR treatment (Figs 4D and EV1A–C).

We next evaluated combination of mCD13-AFR and mCD8α-targeted AcTaferon (mCD8-AFN) therapy in the B16Bl6 mouse melanoma model. mCD8-AFN therapy, which had no effect on itself, clearly synergized with mCD13-AFR, leading to extensive tumor regression (Fig 4E and F). Again, no toxic side-effects were observed in all combinations tested, further underscoring the selectivity and safety of AcTakine therapy.

### IFN-γ sensitizes tumor endothelial cells for TNF activity

The synergistic antitumor effect of TNF and IFN-γ has been extensively documented, but the mechanisms underlying this synergy remain poorly understood. In the B16Bl6 model, this synergism is more clear when mIFN-γ is combined with hTNF (Brouckaert et al, 1986). Human TNF, which can be considered as a low-toxic, fast-cleared mTNF mutant in the mouse (Ameloot et al, 2002), is unable to induce complete tumor regression in the B16Bl6 model, except when combined with mIFN-γ (Fig 5A). To investigate whether the synergistic effect of TNF and IFN-γ depends on direct actions on tumor cells or on host cells, we generated IFN-γ-insensitive B16Bl6 tumor cells (B16-dnIFN-γR). When wt mice carrying a B16-dnIFN-γR tumor were treated with hTNF and mIFN-γ, the synergistic anti-tumor effect was still present (Fig EV2A). By contrast, this synergy was completely absent in IFN-γR knockout mice carrying a parental B16Bl6 tumor (Fig EV2B), pointing toward the critical role of the host microenvironment. We next hypothesized that IFN-γ may also act on endothelial cells, sensitizing them for the effects of TNF. We therefore generated transgenic mice in which the vasculature is selectively unresponsive to IFN-γ, by endothelial-specific Flk1 promoter-driven expression of a truncated, dominant-negative IFN-γR (Fig EV2C–E). As shown in Fig 5B, the treatment of these mice resulted in complete loss of the synergy between hTNF and mIFN-γ, confirming the critical role of IFN-γ signaling in endothelial cells.

We also tested whether IFN-γ could sensitize HUVECs to TNF, using either wtTNF or hCD13-AFR. Indeed, while IFN-γ treatment alone had no effect on IL-8 secretion by HUVECs, it sensitized for TNF signaling leading to a two- to threefold increased IL-8 secretion when compared with wtTNF or hCD13-AFR alone (Fig 5C). Of note, IL-8 secretion levels of hCD13-AFR plus hIFN-γ-stimulated cells were equal to those after stimulation with wtTNF alone. Since both target and receptor expression levels are expected to affect responses to AcTakines, we analyzed the expression levels of CD13 and TNF-R1 after stimulation with IFN-γ via qPCR (Fig EV3). While CD13 expression was not significantly changed, TNF-R1 expression in HUVECs was approx. 50% increased after 2, 6, or 24 h of stimulation with IFN-γ.

### Type II AcTaferon synergizes with CD13-AFR for tumor destruction

Based on the findings above, we next designed a type II AcTaferon (AFN-II) (Fig 5D). Deletion of the 8 C-terminal amino acids reduced

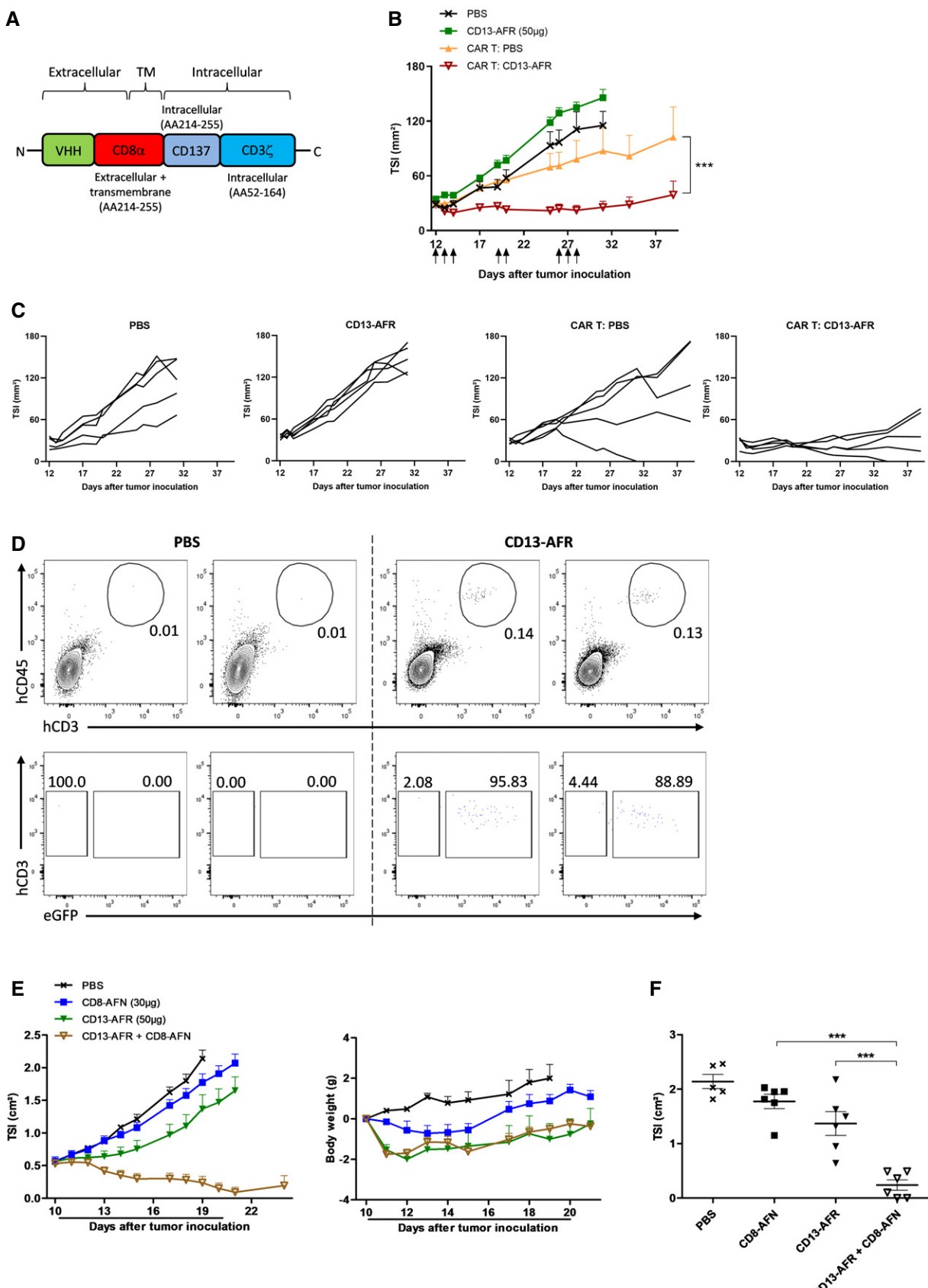

Figure 4.

◄

Figure 4. mCD13-AFR synergizes with immunotherapy.

A    Schematic representation of the CAR construct used, consisting of N-terminal anti-hCD70 VHH, hCD8α hinge and transmembrane (TM) domain, CD137 co-stimulatory domain, and C-terminal CD3ζ activation domain.

B, C  SKOV3 tumor growth in NSG mice after i.v. injection of $6 \times 10^6$ hCD70 CAR T cells on day 12 and p.l. treatment with 50 μg mCD13-AFR (↑). Tumor growth is shown as mean TSI, and error bars are SEM ($n = 5$) (B) or as TSI of individual mice (C). ***$P < 0.001$ by two-way ANOVA with Bonferroni's multiple comparison test.

D    Flow cytometric analysis of hCD45⁺hCD3⁺eGFP⁺ CAR T cells in SKOV3 tumors on day 18 after tumor inoculation. Mice were injected with $6 \times 10^6$ CAR T cells on day 12 and treated with PBS or CD13-AFR (↑ in panel B). Two replicate tumors are shown. See Figure EV1A for gating strategy.

E    B16Bl6 tumor growth and body weight change after daily p.l. injection of 30 μg mCD8-AFN, 50 μg mCD13-AFR, or a combination thereof. Tumor growth is shown as mean TSI + SEM ($n = 6$). The line under the graph represents the treatment period.

F    TSI of individual tumors of the indicated treatment groups at day 19 after tumor inoculation. Error bars are SEM. ***$P < 0.001$ by one-way ANOVA with Bonferroni's multiple comparison test.

Source data are available online for this figure.

mIFN-γ activity approx. 7,000-fold in an encephalomyocarditis virus (EMCV) cytopathic assay in L929 cells (Fig EV4) and more than 3,000-fold in a TNF/IFN-γ cytotoxicity assay in B16Bl6 cells (Fig 5E). As a mCD20-AFN-II, the activity was completely recovered and even surpassed that of wt mIFN-γ on B16Bl6 cells expressing mCD20, predicting a more than 10 000-fold AcTakine targeting efficiency. To target IFN-γ activity to tumor vasculature, we used the same mCD13 VHH as for AFR. In both the B16Bl6 model and the human RL lymphoma model in immunodeficient NSG mice, daily treatment with mCD13-AFN-II resulted in significant tumor growth suppression (Fig 6A and B). To evaluate the *in vivo* synergy between mCD13-AFR and mCD13-AFN-II, we used equimolar doses in combination treatments. In both tumor models, combination therapy induced very rapid tumor necrosis, resulting in complete regression within 6 days of treatment in all mice (Fig 6A and B), demonstrating the generic and direct effect on tumor vasculature, independent of an immune response. The kinetics and appearance of tumor necrosis induced by mCD13-AFN-II/mCD13-AFR combination therapy (Fig EV5) were similar to those observed during treatment with high-dose wtTNF alone, suggesting that mCD13-AFN-II is able to shift mCD13-AFR signaling toward cell death. This was further confirmed by the detection of selective endothelial apoptosis in B16Bl6 tumors as soon as 3–5 h after mCD13-AFN-II/mCD13-AFR treatment (Fig. 6C). Strikingly, even this synergistic mCD13-IFN-II/mCD13-AFR co-treatment did not evoke a detectable toxic response.

## Discussion

Cancer immunotherapy rapidly evolves toward combination treatments to enhance therapeutic efficacy. In this context, it is very unfortunate that the considerable potential of cytokines cannot be fully exploited due to severe toxicity issues. Indeed, to date, only IFNα and IL-2 have found limited clinical use for the treatment of human cancer due to a narrow therapeutic window (Conlon *et al*, 2019). For TNF, this window is even non-existent: The predicted effective dose of TNF is at least 5 mg/m² (~ 120 μg/kg, based on dosing in animal models and ILP patients), exceeding by far the maximum tolerated dose of 150–400 μg/m² (Lejeune *et al*, 2006; Roberts *et al*, 2011). AcTakines provide a solution to the cytokine toxicity problem (Garcin *et al*, 2014). AcTakines are only active on the targeted cell type, allowing the separation of desired from undesired effects and hence safe exploitation of their clinical potential. A key aspect in the design of an AcTakine is the choice

of surface marker on the desired cell type. While it has long been known that TNF exerts its antitumor effect via stromal cells in the tumor microenvironment, we here demonstrated, using conditional TNF-R1 reactivation knockout mice, that expression of TNF-R1 on endothelial cells is sufficient for its antitumor effect. Moreover, we found that endothelial TNF-R1 reactivation knockout mice were completely resistant to acute TNF toxicity, indicating that directing TNF activity only to endothelial cells of the tumor vasculature could generate an effective yet safe therapy. Tumor vasculature-targeted TNF immunocytokines are being developed, most notably NGR-TNF and L19-TNF, which are currently in phase II/III clinical trials for, respectively, mesothelioma and melanoma (Danielli *et al*, 2015a; Gregorc *et al*, 2018). These drugs, however, do not allow full exploitation of TNF's potential due to an inherent risk of side-effects and so-called "sink effect". We applied the AcTakine concept to both mouse and human TNF, resulting in AFRs with a targeting efficiency exceeding 100-fold, and targeted them to CD13 expressed on endothelial cells of the tumor neovasculature. We demonstrate that mAFR is safe in a model for acute TNF toxicity, up to doses surpassing the estimated effective dose of wtTNF more than 50-fold. When targeted to tumor vasculature, mAFR induces clear and prolonged activation of tumor vasculature, evidenced by expression of leukocyte adhesion markers. It is intriguing that mCD13-AFR used as a single agent, in contrast to wtTNF, appears unable to induce disruption of tumor vasculature. This is in accordance with the work of Johansson and colleagues showing that tumor vasculature-targeted TNF (via an RGR peptide) improved tumor vessel function and reduced leakiness in a Rip1-Tag5 pancreatic tumor model (Johansson *et al*, 2012). Also on HUVECs, hCD13-AFR activity appears slightly reduced compared to wtTNF, while this is not the case in assays on other cell types, suggesting that in endothelial cells the signaling cascade initiated by AFR may differ from that of wtTNF. Although our data clearly point toward a crucial role for TNF-R1, a contribution of TNF-R2, which is expressed on endothelial cells and was reported to be important for TNF-induced ICAM-1, VCAM-1, and E-selectin upregulation in the endothelium (Chandrasekharan *et al*, 2007), is likely. Importantly, the difference between wtTNF and CD13-AFR cannot be explained by differential induction of IFN-γ, since mTNF is as efficient in inducing tumor necrosis in IFN-γR⁻/⁻ mice as in wt mice, indicating that its antitumor effect is independent of IFN-γ (Fig EV2B). Taken together, the selective, non-injurious activation of tumor vasculature indicates that CD13-AFR could be an effective drug to improve chemotherapy or immunotherapy.

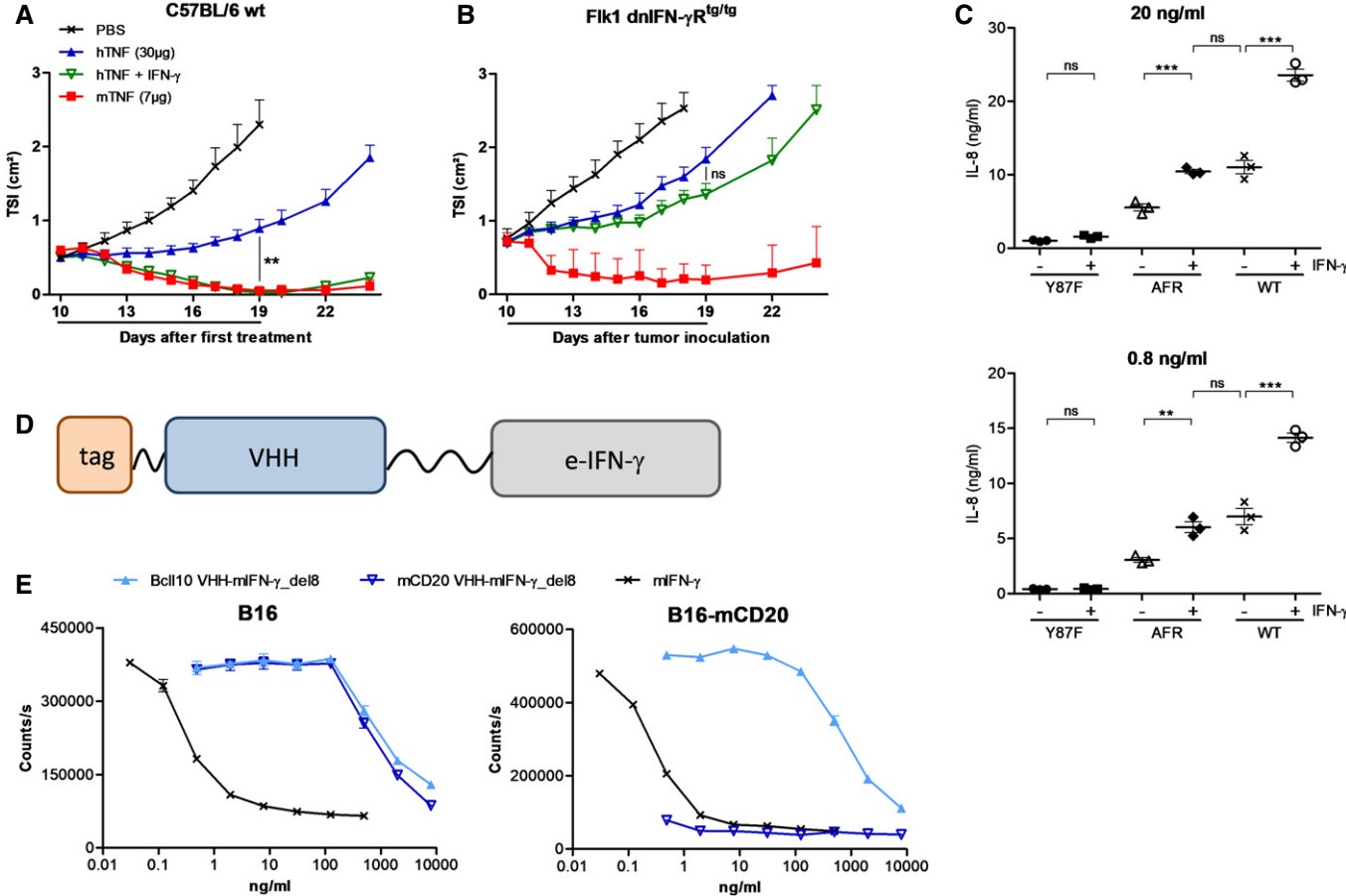

**Figure 5. IFN-γ sensitizes tumor endothelial cells for TNF.**

A, B   B16Bl6 tumor growth after daily p.l. treatment with 7 μg mTNF, 30 μg hTNF alone, or in combination with 10 000 IU mIFN-γ in wt C57BL/6 mice (A) or homozygous Flk1 dnIFN-γR1 transgenic mice (B). The line under the graph represents the treatment period. Tumor growth is depicted as mean TSI, and error bars are SEM (n = 5). ns, non-significant; **P < 0.01 by one-way ANOVA with Bonferroni's multiple comparison test.

C       IL-8 concentration in HUVEC supernatant after 24-h stimulation with the indicated concentration of sc hTNF mutant Y87F (Y87F), hCD13-AFR or wt hTNF, alone or in combination with 200 ng/ml hIFN-γ, measured via ELISA. Error bars are SEM. ns, non-significant; **P < 0.01; ***P < 0.001 by one-way ANOVA with Bonferroni's multiple comparison test. Individual values are shown. The horizontal lines indicate the mean per condition.

D       Schematic representation of AFN-II, consisting of a VHH fused via a 20xGGS linker to an engineered IFN-γ (e-IFN-γ) variant and N-terminal affinity tag.

E       Viability of B16Bl6 parental or mCD20-expressing cells after 72-h stimulation with wt mIFN-γ or del8 mutant fused to a BcII10 or mCD20 VHH, in the presence of 60 ng/ml mTNF. Cell viability was measured via an ATP luminescence assay. Each point is the mean of three replicates, and error bars are SEM.

Source data are available online for this figure.

---

Novel immunotherapies such as DC-based therapies, adoptive T-cell therapies, and therapeutic cancer vaccines all require the penetration and perpetuation of immune cells, especially cytotoxic T cells, in the tumor. One of the main natural functions of TNF is activation of local endothelia to attract circulating immune cells. It is therefore no surprise that TNF can synergize with immunotherapies such as IL-2 (Schwager et al, 2013; Danielli et al, 2015a,b; De Luca et al, 2017) or adoptive T-cell transfer (Johansson et al, 2012). We have previously reported that low-dose TNF therapy potentiated DC-targeted AFN therapy (Cauwels et al, 2018b). Here, we show the remarkable synergy of vasculature-targeted AFR therapy with mCD8-AFN, targeting cytotoxic T cells (and the cDC1 subset of DCs), leading to profound B16Bl6 tumor elimination. Furthermore, we demonstrate that mCD13-AFR therapy potentiates CAR T-cell immunotherapy in a human solid tumor model, resulting in

increased tumor infiltration of CAR T cells and better tumor control. Although more research is needed, our data suggest that vasculature-targeted AFR therapy may be employed to enhance the efficacy of cytotoxic T-cell-based or other immunotherapies.

The synergistic antitumor effect of TNF and IFN-γ has been documented in numerous experimental models and in human ILP patients. We here show that TNF and IFN-γ act on the same endothelial target cell, and thus generated a CD13-targeted IFN-γ AcTakine. Combination therapy of mCD13-AFR and mCD13-AFN-II induced rapid and complete tumor necrosis in two completely different tumor settings. This synergistic effect relied on signaling in endothelial cells and may be explained by AFN-II-induced upregulation of pro-apoptotic genes (e.g., pro-caspase-8 (Li et al, 2002)) or TNF receptors (Wang et al, 2006). Indeed, it was reported before that TNF-R1 expression levels are critical for sensitivity to TNF (Van

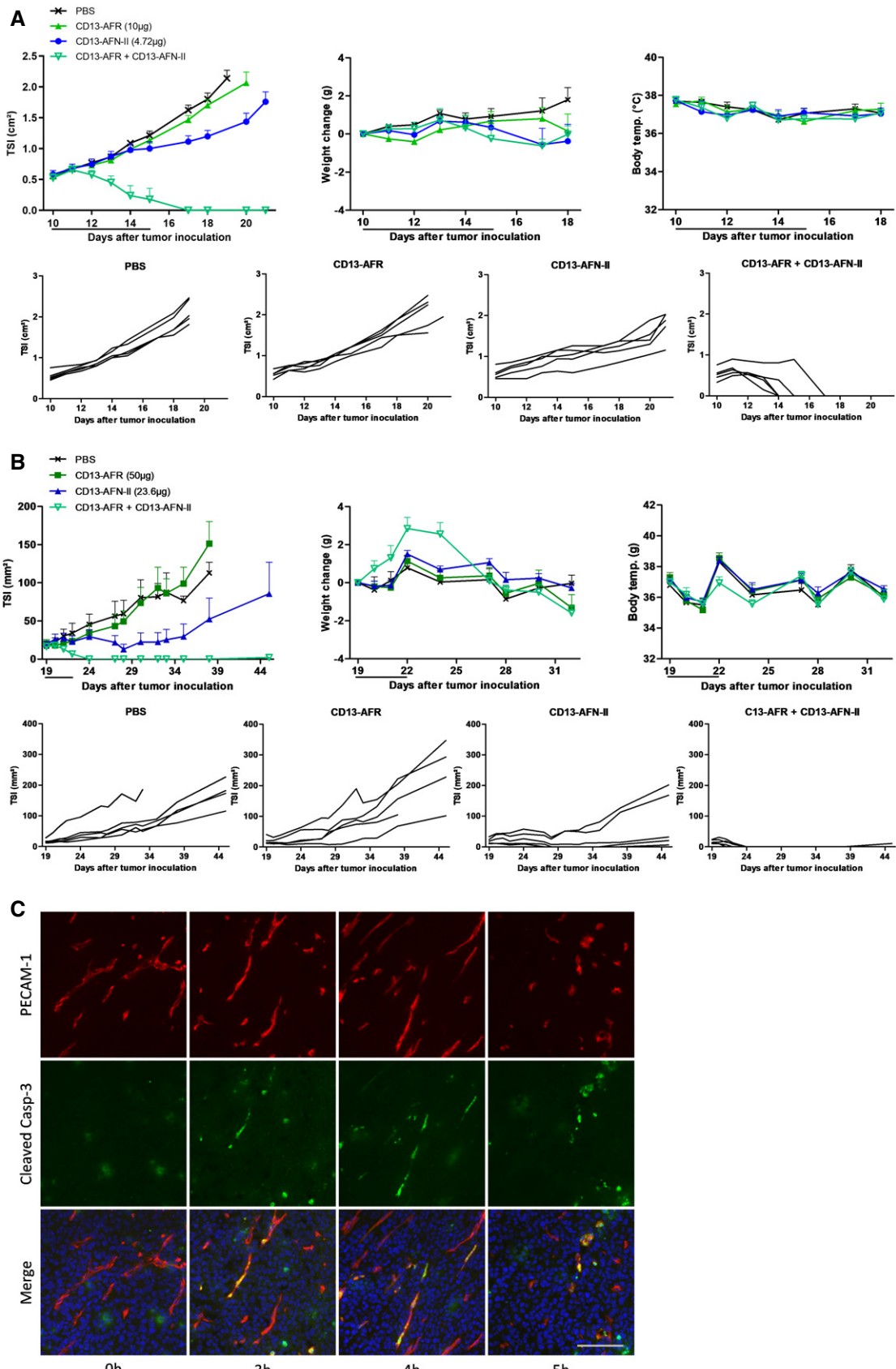

**Figure 6.**

**Figure 6.  CD13-AFN-II synergizes with CD13-AFR for tumor destruction.**

A, B    Tumor growth, body weight change, and rectal body temperature after daily p.l. treatment with mCD13-AFR, mCD13-AFN-II, or combination thereof, in the B16Bl6 model in C57BL/6 mice (A) or RL model in NSG mice (B). Tumor growth is shown as mean TSI, and error bars are SEM (*n* = 5) (upper panel), or as TSI of individual mice (lower panel). The line under the graph represents the treatment period.

C       Immunohistochemical staining for PECAM-1 (red), cleaved caspase-3 (green), and DNA (blue) of B16Bl6 tumors after p.l. treatment with 50 μg mCD13-AFR and 23.8 μg mCD13-AFN-II. Scale bar is 100 μm.

Source data are available online for this figure.

Hauwermeiren *et al*, 2013), and IFN-γ upregulated TNF-R1 on HUVECs. The fact that we observed caspase-3 activation in tumor vasculature suggests that AFR/AFN-II therapy induced endothelial apoptosis, leading to tumor ischemia and necrosis. In addition, Kammertoens *et al* (2017) have recently reported that IFN-γ may also exert direct anti-vascular effects (Johansson *et al*, 2012) and accordingly, mCD13-AFN-II monotherapy caused significant tumor growth suppression in the RL model. Since we did not observe any detectable toxicity in the AFR and AFN-II treatments, our strategy may offer a more efficacious and safer alternative for existing anti-angiogenic therapies such as VEGF pathway blockade (Chen & Cleck, 2009).

While CD13-AFR and CD13-AFN-II were designed with clinical translatability in mind, it is important to note that determinants for clinical safety and efficacy, such as immunogenicity and pharmacokinetics, were not addressed in this study. Also, the subcutaneous, fast-growing tumor models used in this work might not fully reflect human cancers. Additional work in more relevant models, such as genetic mouse models or PDX models, and with systemic delivery methods will be required to further develop this new class of biologics. To conclude, the work here presented provides a first proof of concept for two novel, versatile protein drugs designed to selectively target TNF and IFN-γ activity to the tumor vasculature and which in different combination modalities can lead to eradication of large established tumors without toxicity. As no tumor markers are needed, safe elimination of a broad range of tumor types may become feasible.

## Materials and Methods

### Reagents and cells

mTNF, hTNF, and mIFN-γ were produced by VIB Protein Service Facility (Dr. J. Haustraete). mIFN-γ had a specific activity of $1.16 \times 10^8$ IU/mg. hIFN-γ was purchased from PeproTech. Wortmannin was purchased from Sigma-Adrich, dissolved in DMSO at 1 mg/ml, and further diluted in PBS. Birinapant was purchased from LC Laboratories, dissolved in DMSO at 50 mg/ml, and diluted in PBS with 10% DMSO and 2% Tween-80.

B16Bl6 cells were obtained via Dr. M. Mareel (Ghent University Hospital) (Poste *et al*, 1980). L929, MCF7, RL, and SKOV3 cells were purchased from ATCC. Cell lines were cultured in RPMI (RL) or DMEM (other) with 4.5 g/l D-glucose, L-glutamine, and pyruvate (Gibco), supplemented with 10% FCS. FreeStyle 293-F cells (Gibco) were purchased from Thermo Fisher Scientific and cultured in Free-Style 293 Expression Medium (Gibco). Cell lines were routinely tested for mycoplasma contamination via Venor GeM Classic mycoplasma PCR detection kit (Minerva Biolabs).

To generate B16-dnTNF-R1 cells, the extracellular and transmembrane part of mTNF-R1 from pBluMTNF-R55 (Dr. W. De Clercq) and the intracellular part of the dominant-negative hTNF-R1 and C-terminal E-tag from pCDNA1 hTNF-RIdel243-383 (Boone *et al*, 2000) were ligated in pCAGGS resulting in the vector pCAGGs-TNF-RI del IC dn. This vector was transfected with Lipofectamine (Invitrogen) in B16Bl6 cells. Individual clones were generated by limiting dilution and screened for dnTNF-R1 expression via E-tag.

To generate B16-dnIFNγ R cells, the dnIFN-γRα chain from p1017.myc-mgR del IC (Dighe *et al*, 1995) was cloned into pGEMT, generating pGEMT-gR del IC-myc. The dnIFN-γRα chain was further cloned into pCAGGS (Dr. J. Miyazaki, University of Tokyo, Japan) after the strong modified β-actin promoter, resulting in pCAGGS-gR del IC-myc. This construct was co-transfected with pBSpac delp (Dr. J. Ortir, Universidad Autonoma, Madrid, Spain) in B16Bl6 cells with Lipofectamine Plus (Invitrogen). Puromycin-resistant cells were cloned by limiting dilution method, and individual clones were screened for dnIFN-γR expression (Myc-tag) and unresponsiveness to IFN-γ.

Human umbilical vein endothelial cells were isolated from fresh umbilical cords collected at AZ Sint-Lucas (Ghent) via collagenase IV (Sigma) digestion and cultured in complete EGM-2 medium (Lonza) without hydrocortisone. All experiments were performed with cells at passage numbers 3–5.

### Construction and production of AFR, AFN-II, & AFN

VHHs were generated by VIB Nanobody Core (Dr. G. Hassanzadeh). Selection of CD8α, CD13, and CD20 VHHs was based on binding to HEK293T cells transiently transfected with the corresponding proteins, binding to primary cells (HUVEC for hCD13, mouse splenocytes for mCD8α and mCD20), and, in case of CD13, on inhibition of enzymatic activity with L-Leucine-*p*-nitroanilide (Sigma-Aldrich) as substrate. scTNF was generated by C- to N-terminal fusion of 3 TNF monomers via a GGGGS linker. Mutations in scTNF and mIFN-γ were introduced via site-directed mutagenesis. AFRs consist of N-terminal VHH followed by 20xGGS linker, mutated scTNF, and C-terminal 6xHis-tag. AFN-IIs consist of N-terminal 6xHis-tag followed by VHH, 20xGGS, and mutated mIFN-γ. AFRs and AFN-IIs were cloned in a mammalian expression vector, based on pmKate2-N (Evrogen), containing the CMV-IE promoter, SV40 poly A signal, and SIgK leader sequence. Large-scale productions were performed in FreeStyle 293-F cells (Gibco). Cells were transfected using PEI (Polysciences) and co-transfected (1:50 ratio) with a vector encoding SV40 large T antigen. After 5 days, supernatant was collected, filtered through a 0.2-μg PES membrane filter, and His-tagged protein was purified by immobilized metal ion chromatography on Ni Sepharose Excel (GE Healthcare) resin. Columns were washed with a buffer

containing 30 mM imidazole (Merck), and protein was eluted with 300 mM imidazole. Imidazole was removed by gel filtration over PD-10 desalting columns (GE Healthcare). Protein concentration was measured via absorbance at 280 nm (NanoDrop), and purity was assessed via SDS–PAGE.

mCD8-AFN and mClec9a-AFN were generated and produced in E. coli as previously described (Cauwels et al, 2018b).

### In vitro bioactivity assays

For cytotoxicity assays, MCF7, L929, or B16Bl6 cells were plated in black 96-well tissue culture plates (Nunc) at 1,000 cells/well 24 h before stimulation. After 72 h of stimulation, cell supernatant was removed and cell viability was determined with CellTiter-Glo (Promega).

For the cytopathic effect assay, L929 cells were plated in black 96-well tissue culture plates (Nunc) at $8 \times 10^4$ cells/well in DMEM with 2% FCS. The next day, mIFN-γ or mAFN-II was added and cells were incubated overnight. EMC virus (ATCC) was added, and 24 h later, cell viability was determined with CellTiter-Glo (Promega).

To assay IL-8 secretion by HUVECs, cells were plated at $2 \times 10^4$ cells/well in 96-well plates the day before stimulation. After 24 h of stimulation, supernatant was collected and IL-8 concentration was determined via Human IL-8 ELISA MAX kit (BioLegend).

### Animal studies

C57BL/6J and NSG mice were purchased from Charles River Laboratories (France). TNF-R1$^{-/-}$ mice were generated by Dr. H. Bluethmann (Roche Research, Basel, Switzerland) (Rothe et al, 1993). p55$^{cneo/cneo}$ mice were generated by Dr. G. Kollias (Institute of Immunology Biomedical Sciences Research Center "Al. Fleming", Vari, Greece) (Victoratos et al, 2006). Flk1Cre mice were obtained from Dr. G. Breier (Max Planck Institute for Physiological and Clinical Research, Bad Nauheim, Germany) (Licht et al, 2004), DelCre mice from Dr. W. Müller (Institute for Genetics, University of Cologne, Germany) (Betz et al, 1996). Cre mice were crossed with p55$^{cneo/cneo}$ mice until homozygosity of the TNF-R1$^{cneo}$ allele. The Cre transgene was kept hemizygous. Mice were genotyped by PCR with primers 5′-TGG TGG CCT TAA ACC GAT CC-3′, 5′-AGA GAG GTT GCT CAG TGT GAG GC-3′ and 5′-ATG ATT GAA CAA GAT GGA TTG CAC-3′. For LEC isolation, a single-cell suspension of lung was made by collagenase A (Sigma) treatment. Cells were stained with anti-mCD31-PE Ab (BD Pharmingen, clone MEC13.3, diluted 1/50), and CD31$^+$ LECs were MACS-sorted with anti-PE microbeads (Miltenyi Biotec).

To generate Flk1 dnIFNγR$^{tg/tg}$ mice, the dnIFN-γRα chain from pGEMT-gR del IC-myc was ligated in pGLacZ-Flk1 (Kappel et al, 1999). The plasmid backbone was removed by SalI/XmaI digestion, and the remaining 4.5-kb fragment was microinjected into fertilized mouse oocytes. Generation of transgenic mice was performed as described previously (Kappel et al, 1999). Founder lines were genotyped by nested PCR with primer pairs 5′-TCT TTC TGC CCT GAG TCC TC-3′ and 5′-CTT CAG GGT GAA ATA CGA GG-3′, and 5′-CGC CTC TGT GAC TTC TTT GC-3′ and 5′-GAT GCT GTC TGC GAA GGT CG-3′. Founder lines were screened for endothelial-specific expression (Myc-tag) on E10.5.

For tumor experiments, mice were inoculated with $6 \times 10^5$ B16Bl6 cells, $2 \times 10^6$ RL cells, or $2 \times 10^6$ SKOV3 cells, s.c. in the shaved back. Treatment was started at least 10 days after tumor inoculation. Tumor size index (TSI), the product of the largest perpendicular diameters, was measured with a caliper. Unless otherwise mentioned, mice were treated daily via p.l. injection of 100 μl with a 30G insulin syringe. For qPCR analysis and toxicity studies, mice were injected with 200 μl i.v. using a 30G insulin syringe. Blood samples were collected via the lateral tail vein in Microvette K3 EDTA tubes (Sarstedt). Plasma was separated by centrifugation at 1,500 g for 10′, and IL-6 concentration was determined via Mouse IL-6 ELISA MAX kit (BioLegend).

### qPCR analysis

Snap-frozen B16Bl6 tumor samples were disrupted and homogenized with a TissueLyser LT (Qiagen) machine, and RNA was isolated with the RNeasy Mini Kit (Qiagen), including DNaseI treatment. cDNA synthesis was performed with the Biotechrabbit cDNA Synthesis Kit, using both random hexamer primers and oligo(dT) primers. qPCR analysis was done with LightCycler 480 SYBR Green I Master and LightCycler 480 instrument (Roche). Reference genes were selected from a panel of 8 with Genorm (Dr. J. Vandesompele, UGent), and normalized gene expression was calculated with the delta CT method (Vandesompele et al, 2002). Primers used were 5′-TCA CCG CTT CAG AAA ACC ACC-3′ and 5′-GGT CCA CTG TGC AAG AAG AGA-3′ for human TNF-R1, 5′-TTC AAC ATC ACG CTT ATC CAC C-3′ and 5′-AGT CGA ACT CAC TGA CAA TGA AG-3′ for human CD13, 5′-TGC CTC TGA AGC TCG GAT ATA C-3′ and 5′-TCT GTC GAA CTC CTC AGT CAC-3′ for mouse ICAM-1, 5′-ATG CCT CGC GCT TTC TCT C-3′ and 5′-GTA GTC CCG CTG ACA GTA TGC-3′ for mouse E-selectin, and 5′-GGG TCG ATT TCA AAC TCA ATG T-3′ and 5′-AGA GTA AAG CCT ATC TCG CTG T-3′ for mouse VEGF-R2.

### IHC analysis

Samples were frozen in PolyFreeze tissue freezing medium (Polysciences) on dry ice. Ten-micrometer cryosections were cut, air-dried for 30′, fixed with 4% PFA for 15′, permeabilized with 0.5% Triton X-100 for 3′, and stained according to standard IHC protocols. The following primary antibodies were used: c-Myc (ATCC, clone 9E10), mouse CD31 (BD Pharmingen, clone MEC13.3), mouse CD45 (BioLegend, clone 30-F11), polyclonal goat anti-mouse ICAM-1 (NovusBio), and cleaved caspase-3 (Asp175) (CST, clone 5A1E), all diluted to a working concentration of 5 μg/ml. Alexa Fluor 488-, Alexa Fluor 568-, or Alexa Fluor 594-labeled secondary antibodies raised in donkey (Invitrogen) were diluted 1/200. Confocal images were taken with an Olympus FluoView FV1000 microscope.

### CAR T-cell model

Peripheral blood mononuclear cells (PBMCs) were isolated from buffy coat from an anonymous donor (Red Cross Flanders) using Lymphoprep (Stemcell Technologies) and stimulated with ImmunoCult human CD3/CD28/CD2 T-cell Activator (Stemcell Technologies) for selective activation and subsequent proliferation of T lymphocytes (day 0). Three and 4 days later, cell suspensions were

## The paper explained

### Problem

Systemic toxicity still prevents full clinical application of cytokines such as TNF and interferons (IFNs), which hold great potential for cancer immunotherapy. One strategy to reduce cytokine systemic toxicity is immunocytokines, cytokines fused with a targeting antibody. Although immunocytokines display an increased activity on target cells, off-target side-effects are expected to remain a major obstacle.

### Results

We have developed a new class of immunocytokines, named AcTakines (Activity-on-Target cytokines), that allows selective targeting of cytokine activity to one specific cell type. In this study, we have identified the tumor endothelium as the target cell for the antitumor effect of TNF and IFN-γ. AcTakines were generated by fusing an inactivated TNF or IFN-γ mutein with a tumor vasculature-targeting CD13 VHH (resulting in CD13-AFR and CD13-AFN-II, respectively). Treatment with CD13-AFR enabled selective activation of the tumor vasculature without toxicity. This supported enhanced immune cell infiltration in the tumor leading to better control or elimination of solid tumors by, respectively, CAR T cells or a combination therapy with a CD8-targeted IFN-α AcTakine. Co-treatment of CD13-AFR and CD13-AFN-II resulted in very rapid destruction of the tumor vasculature and complete regression of large tumors, without any detectable toxicity.

### Impact

Our results indicate that CD13-AFR may open new therapeutic opportunities, (i) by boosting the efficacy of existing immunotherapies, or (ii) in combination with CD13-AFN-II as alternative for toxic vascular-disrupting agents (VDAs) currently used in the clinic. As no tumor-specific markers are needed, safe and efficacious elimination of a broad range of tumor types may become feasible.

retrovirally transduced with the LZRS-IRES2-EGFP (LIE) vector encoding the chimeric antigen receptor (CAR) sequence. The expressed CAR is composed of an anti-hCD70 VHH, a CD8α-based hinge, the co-stimulatory domain of 4-1BB (CD137), and the T-cell receptor-derived signaling domain CD3ζ. Six days after the second transduction (day 10), T cells were functionally analyzed by measuring intracellular IL-2 and IFN-γ expression using the Cytofix/Cytoperm kit (BD Biosciences) and flow cytometry. At day 14, CAR T cells were harvested, washed using sterile PBS, and diluted in PBS for injection in mice.

NSG mice were s.c. injected with $2 \times 10^6$ RL cells or $2 \times 10^6$ SKOV3 cells. When tumors reached a size of 4–7 mm in diameter, mice were injected i.v. with PBS or $2.8 \times 10^6$ to $8 \times 10^6$ CAR T cells. That same day, 2 h after CAR T-cell injection, treatment with PBS or CD13-AFR was started via p.l. injection. Tumor-free mice were checked for the presence of remaining tumor cells with a luciferase assay, in which mice were injected i.p. with 150 μg/g XenoLight D-luciferin (PerkinElmer). After 10-min incubation, luciferase activity was visualized using a Lumina II *in vivo* imaging system (IVIS; PerkinElmer).

For flow cytometric analysis of CAR T cells in tumors, mice were sacrificed. SKOV3 tumors were excised and grinded to single-cell suspensions using a 5-ml syringe plunger. Cells were resuspended in ACK lysing buffer to remove red blood cells and incubated with anti-mouse CD16/32 (BioLegend) and human Fc blocking buffer (Miltenyi) for 5 min on ice, followed by propidium iodide

(Invitrogen), anti-hCD3-PeCy7 (eBioscience; clone UCHT1), and anti-hCD45-BV510 (BD Biosciences; clone HI30) antibodies (diluted 1/100) for 30 min on ice. Cells were washed with PBS and analyzed on a BD LSR II flow cytometer (BD Biosciences). Viable hCD45+hCD3+CAR T cells were gated based on morphology and granularity (SSC vs. FSC), viability (propidium iodide, negative), and exclusion of doublet cells (FSC-width vs. FSC-area).

## Study approval

All animal experiments followed the Federation of European Laboratory Animal Science Association guidelines and were approved by the Ethical Committee of Ghent University. Isolation of HUVECs and experiments with primary human cells were approved by the Medical Ethics Committee of Ghent University Hospital. An informed consent was obtained from all donors. All experiments with human materials conformed to the principles set out in the WMA Declaration of Helsinki.

## Data analysis and statistics

Data were analyzed with GraphPad Prism software. For animal studies, groups were stratified based on TSI (B16Bl6) or randomized (RL and SKOV3) before the first treatment. Statistical differences between groups or conditions were determined by one-way or two-way ANOVA, followed by Bonferroni *post hoc* test. Exact *P*-values are listed in Appendix Table S1.

**Expanded View** for this article is available online.

## Acknowledgements

We would like to dedicate this article to Walter Fiers (1931–2019). We thank Reza Hassanzadeh Ghassabeh (VIB Nanobody Core) for the generation and selection of VHHs; Jürgen Haustraete (VIB PSF) for help with AcTakine productions in HekF cells; Elke Rogge, Dominiek Catteeuw, and Jennifer De Geest for excellent technical support; George Kollias and Tino Hochepied for the p55$^{cneo/cneo}$ mice and Flk1 dnIFN-γR transgenic mice, respectively; and AZ Sint-Lucas Gent and Red Cross Flanders for umbilical cords and buffy coats, respectively. This work was supported by UGent Methusalem, ERC Advanced (CYRE, No. 340941), and Proof of Concept (AcTafactors, No. 680889) and FWO-V G009614N grants; grants from LabEx MAbImprove, Institut Carnot CALYM, the Canceropôle—Institut National du Cancer (INCa) to GU; and by Orionis Biosciences.

## Author contributions

LH and JT conceived and designed the research. BV and PB designed the CAR T and mutant mice experiments. LH, FP, AVP, AC, SVL, SDM, LZ, and JB aided and conducted the AcTakine optimizations and mouse experiments, including the CAR T work. AG and JH generated and initiated the work with the TNF-R1 and IFN-γR1 mutant mice and B16 cells. NV and AV provided technical support, including VHH selection, constructions, and purifications. GU and NK helped with experimental designs. LH and JT wrote the manuscript.

## Conflict of interest

JT and NK are affiliated with Orionis Biosciences (as scientific advisor and/or employee) and hold equity interests in Orionis Biosciences. JT received financial research support from Orionis Biosciences NV. All the other authors declare that they have no conflict of interest.

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
