## [Review Process File · EMBO Molecular Medicine]

Safe eradication of large established tumors using neovasculature-targeted Tumor Necrosis Factor-based therapies

Leander Huyghe, Alexander Van Parys, Anje Cauwels, Sandra Van Lint, Stijn De Munter, Jennyfer Bultinck, Lennart Zabeau, Jeroen Hostens, An Goethals, Nele Vanderroost, Annick Verhee, Gilles Uzé, Niko Kley, Frank Peelman, Bart Vandekerckhove, Peter Brouckaert & Jan Tavernier

Review timeline:

Submission date:	29 July 2019
Editorial Decision:	14 August 2019
Revision received:	31 October 2019
Editorial Decision:	20 November 2019
Additional correspondence:	21 November 2019
Revision received:	29 November 2019
Accepted:	4 December 2019

Editor: Lise Roth

Transaction Report:

1st Editorial Decision

14 August 2019

Thank you for the submission of your manuscript to EMBO Molecular Medicine. We have now received feedback from the three reviewers who agreed to evaluate your manuscript. As you will see from the reports below, the referees acknowledge the interest of the study and are overall supporting publication of your work pending adequate revisions.

Addressing the reviewers' concerns in full will be necessary for further considering the manuscript in our journal, and acceptance of the manuscript will entail a second round of review. EMBO Molecular Medicine encourages a single round of revision only and therefore, acceptance or rejection of the manuscript will depend on the completeness of your responses included in the next, final version of the manuscript. For this reason, and to save you from any frustrations in the end, I would strongly advise against returning an incomplete revision.

I look forward to receiving your revised manuscript.

***** Reviewer's comments *****

Referee #1 (Comments on Novelty/Model System for Author):

The study is based on sophisticated animal models allowing systematic and mechanistic testing of hypotheses in in vivo conditions.

The approach of reducing the binding affinity of a cytokine towards its receptor and combining it with a targeting moiety to enhance ligand-receptor interaction and thus restore high biological activity deserves more attention from the scientist community.

Ethical concerns: The study contains a substantial number of experiments where 100% lethal dose was determined for various TNF variants. It was not necessary to make the point that the developed mutant is not toxic. This should have been avoided.

While I am not an ethicist, it should be brought to the attention of the authors.

Referee #1 (Remarks for Author):

The manuscript by Huyghe and colleagues describes the development and pre-clinical potential of low-affinity, targeted cytokine-based therapeutics against certain solid tumours. The manuscript is very clearly written and the approaches and results are potentially very significant.

I have 3 main comments regarding the work:

1. It would be important to corroborate the claim that the Y86F TNF mutant upon fusion to a targeting antibody regains its adequate biological activity. Assays, more directly probing TNFR1 activation, such as TNFR1 complex(s) formation and characterisation of key molecular components of the complex in parallel with that of induced by WT TNF would confirm that the mutant activate native TNF signalling.
2. While it is remarkable that a mutant with a 10,000 fold lower affinity against its target can produce nearly maximal TNFR1 activation, it would be worthwhile to look more into the limitations of such mutants. Most importantly, how would the function of the Y86F mutant compare with WT TNF in cells expressing low vs high levels of TNFR1? Is there a threshold TNFR1 expression required to induce the biological response?
3. Regarding the in vivo assays, while the combination treatments achieved highly synergistic effect, these experiments lack an essential negative control, which is testing the effect of the targeting unit of the construct, namely CD13, CD20. Targeting CD20 has been reported to induce T cell memory and block Tregs (<https://www.ncbi.nlm.nih.gov/pubmed/25231744>) and it is expressed in immune cell populations, plays a role in B cell activation, etc. Similarly, CD13 is an aminopeptidase, with reported functions of inhibiting IL-8-mediated cell death, etc. CD13 is also highly expressed in ovarian cancer cells, thus the effect of the targeting antibodies themselves on tumour eradication (e.g. via ADCC) would need to be evaluated.

Referee #2 (Comments on Novelty/Model System for Author):

The authors have used a novel conditional endothelial knockout mouse model, elegant AcTakine constructs, as well as B16Bl6 tumor cells lacking TNF-R1 or IFN-gamma. All of these unique reagents allowed for a novel and clear research design appropriate to test the main hypothesis.

Referee #2 (Remarks for Author):

Despite of its potent anti-tumor activity, TNF cannot be used therapeutically in view of its extensive systemic toxicity, characterized by hypotension and hepatotoxicity. Apart from the isolated limb perfusion procedure for the treatment of melanoma and soft tissue sarcoma, developed by the teams of Drs Lejeune and Eggermont, the authors have developed an alternative and elegant approach to target the tumor vasculature in melanoma, while reducing systemic toxicity. They make use of perilesional treatment with an Activity-on-Target cytokine (AcTakine) in a mouse B16Bl6 melanoma model. The AcTakine consists of a single chain mutant Y86F TNF construct, with strongly reduced affinity for TNF receptor 1, fused to an N-terminal VHH against mouse CD13 as a targeting moiety (AFR). This construct prevents unfavorable side and sink effects.

The authors convincingly demonstrate that this AFR AcTakine has an anti-tumor activity towards B16Bl6 melanoma comparable to wt TNF in conditional endothelial TNF-R1 reactivation mice. In WT mice carrying a TNF-unresponsive B16Bl6 tumor treated with TNF, the antitumor effect was similar to that observed with a parental B16Bl6 tumor, thus stressing the importance of TNF-R1 expression in the host cells. Importantly, the recorded systemic toxicity was significantly lower. The authors moreover demonstrate the combined efficacy of tumor vasculature-targeted TNF and IFN-gamma AcTakines to selectively activate or kill tumor endothelial cells. They moreover show that tumor vasculature-targeted AFR therapy synergized with either mouse CD8-AFN immunotherapy, human CAR T-cell immunotherapy and tumor vasculature-targeted AFN-II therapy. This combination caused a complete eradication of large established tumors in mice without significant toxicity.

In general, this important and high impact study uses state of the art technology, including novel TNF constructs novel conditional TNF-R1 reactivation knockout mice that specifically express TNF-R1 in endothelium, and tumor cells lacking cytokines or their receptors. In general, results are clear and support the main conclusions. I only have some minor remarks with regard to the connection between the in vivo and in vitro studies.

1. Why were HUVEC cells used as an endothelial cell model in Fig. 2d? Tumor-Associated ECs are mostly microvascular. MVEC have a significantly different phenotype from large vessel endothelial cells with regard to reactivity to LPS and are moreover more sensitive to TNF-induced cell death. This is moreover indicated by the significantly lower IL-8 potential of the CD13AFR in HUVEC cells (Fig 2d), as compared to wt TNF, but a similar induction of ICAM-1 in the tumor microvasculature (Fig. 3c,d). As such, in vitro data could be included with commercially available human lung microvascular endothelial cells in order to compare the pro-inflammatory and cytotoxic activities of the constructs with wt TNF in these more relevant endothelial population. Results with mouse MVEC would moreover provide a better connection between the in vivo and the in vitro results.

2. Although results clearly point towards a crucial role for TNF-R1 in the in vivo studies, the role of TNF-R2 is kind of ignored. Yet, despite of not having a death domain, this receptor is important for TNF-induced ICAM-1, VCAM-1 and E-selectin upregulation in endothelium (Chandrasekharan et al., Blood, 2007) and, upon co-activation, can increase cell death induction by TNF-R1, as was also shown in mouse and brain MVEC. The use of hTNF in mice, which does not activate mouse TNF-R2, does not allow this distinction either. Does the Y86F mouse TNF mutant also have reduced binding to TNF-R2?

3. Fig S1 demonstrating equal anti-tumor activity of TNF in mice with a wt B16B16 or a TNFR1^{-/-} tumor should be included in the main manuscript since these are important data.

4. It is recognized that in Fig. 3b, the sc mutant TNF construct does not induce systemic toxicity in vivo, but it also does not induce any IL-6, as compared to controls (the latter of which have rather high basal IL-6 levels in plasma). As such, in this experiment there is no proof that the sc mutant TNF construct has any activity at all.

5. The coupling of Y86F mTNF and Y87F hTNF to VHH mCD20 or hCD20 induces comparable cytotoxicity as wt TNF in L929 and MCF-7 cells only if these cells express CD20. Yet, in endothelial cells the presence of CD13 does not allow apoptosis induction by the AFR constructs, neither in vitro nor in vivo, unless when combined with IFN-gamma treatment. Could the expression level of CD13 affect this? Would over-expression of CD13 in EC induce cytotoxicity by the constructs? The authors should discuss this.

6. How do the authors explain the anti-tumor effect of the AFR in the B16B16 melanoma model (Fig. 3e), despite of the fact that the construct does not induce endothelial apoptosis?

7. CD13 was recently shown to be expressed not only in tumor vasculature but also in tumor cells, e.g. in patients with soft tissue sarcoma (Kessler et al., Transl. Oncol. 2018). As such, could this complicate this approach or confound interpretation of results?

Referee #3 (Comments on Novelty/Model System for Author):

The quality of experiments is good, the novelty medium to high, the impact high. While the mouse models for evaluating toxicity vascular targeting are good, the transplantable tumor models (B16 and the xenogenic tumor) are problematic. The major problem is that the TNF Actikine targets endothelial cells and both transplantable models are very artificial with respect the tumor stroma and building up new vasculature. I think the authors should discuss this limitation critically and also discuss the potential mechanisms for their findings. Currently the manuscript leaves the impression of "marketing" Actikines in various combinations. This can in my opinion be overcome by a more detailed and critical discussion of results.

Referee #3 (Remarks for Author):

The study by Huyghe et al. describes an intriguing approach to reduce systemic toxicity, namely by lowering affinity to the cytokine receptor by altering the amino acid sequence and at the same time by targeting the cytokine to a particular cell type or tissue using antibody fragments, they increase the local concentration and thus compensate for lower receptor affinity by increasing avidity of binding in the target tissue.

Expanding on earlier work (using this strategy for type 1 Interferon), in this study they present data for mutated TNF (mTNF) in various combinations of immunotherapy (mTNF alone, mTNF + adoptive CAR T cell therapy, mTNF + IFN γ , mTNF + type 1 Interferon). They use elegant mouse models with tissue specific TNFR1 expression and down modulation of IFN γ R expression to show that progression of transplanted tumor cell lines can be slowed down or tumor cell inocula can even be eradicated without overt toxicity and that TNF action on the CD13 expressing neovasculature of the tumors is important in their model.

While I think the approach of modifying cytokines, as for example the group of Christopher Garcia has shown earlier this year (creating a biased agonist of IFN γ , see Nature 2019), is topical and specifically using a modified TNF in different combinations of immunotherapy is compelling and in principle suitable for publication in EMBO Molecular Medicine, the manuscript in its current form in my opinion is not yet suitable for publication and needs major changes.

Major concern:

My major concern is that while the strategy is interesting and the mouse model for evaluating host effects (toxicity - targeting neovasculature) are appropriate, the transplantable tumor models (B16 and the xenogenic tumor) are problematic. The major problem is that the TNF Actakine approach targets, a stroma cell population (endothelial cells) and both models are very artificial with respect to the generation of the tumor stroma.

B16 was serially transplanted several hundred times before it became widely available. These types of tumors recruit their stroma in a matter of days, rather than month or years and as stated in Chapter 47 of Fundamental Immunology ("Cancer Immunology", 7th edition. Lippincott-Williams & Wilkins, Philadelphia, PA. pages 1200-1234.) "many of these tumors no longer resemble primary mouse or human tumors that virtually always grow at much slower rates". B16 growth is easily affected by so many variables, e.g. another Belgian group has recently shown that just by feeding a high salt diet, growth of this tumor is slowed down (Willebrand et al. 2019 Frontiers in Immunology).

The Skov3 xenogenic model suffers from a similar limitation (poor model for microenvironment). The complex heterotypic interactions between tumor and stroma cannot be properly evaluated, because some soluble mediators and surface molecules act across the species barrier while others do not. So again the predictive value is limited.

I think it is clearly beyond the scope of this study (and would be unfair) to ask for data in spontaneous tumor models, but the way the results are presented is not rigorous and critical enough. While transplantable models have their value (as demonstrated by the work of Jim Allison), the authors clearly should discuss more critically the limitations for their work and the preclinical model they used.

Minor points:

- Results: In vivo Actakines are used daily perilesional for up to 10 days. Did they test immunogenicity of their mutant cytokine fusion proteins? Since feasibility and safety to efficiently eradicate tumors with this treatment is proposed, immunogenicity of the mutated therapeutic proteins needs to be addressed, if not experimentally, it needs to be at least mentioned in the discussion whether it can be ruled out or not.

- Results: Figure 1, would it not be more accurate to plot temperature graphs as lines of individual mice, instead of substituting 20 degrees for dead mice?

- Results: Figure legends, in general it is difficult to follow group size (e.g. 4 to 10 per group is too broad) and the number of biological replications of in vivo experiments.

- Results: The FLK (VEGFR2) promoter has been described to be active in endothelial cells. However work by the group of Thomas A. Sacke (and others) has also reported activity in other mesodermal cells (e.g. hematopoietic cells and myocytes, see Motoike T. et al. *Genesis*. 2003). Indeed, when we tested the FLK-cre mice from Georg Breier for recombination in Rosa 26 reporter mice, we found recombination in hematopoietic cells. While Figure 1b suggests that in FLK1-Cre x p55cneo/cneo mice, there is no recombination in spleenocytes, for the novel transgenic mice with FLK1 driven dominant-negative IFN γ R, proof of endothelial specific transgene expression in adult mice is lacking.

- Results: Aminopeptidase N (CD13) is expressed on several other tissues besides vascular endothelial cells (such as macrophages, fibroblasts or epithelial cells, see Look A. et al. *JCI* 1989) and there are also more recent reports of soluble CD13. Therefore, when the authors state that "CD13 enables selective targeting to the tumor vasculature", have they ruled out recognition of other cell types. While they exclude weight loss and overt/apparent toxicity, other on target toxicity or side effects cannot be excluded formally if it is not specifically analyzed. Please discuss.

- Results: the p55cneo/cneo model is a very elegant genetic tool and very well used in this study. But it may need an additional cartoon in figure 1b, for better comprehension to the reader that is less familiar with complex compound mutant mouse lines. The term "conditional TNFR1 reactivation knockout" may describe it but seems not self-explanatory.

- Results: Page 5, Acatfactors,we evaluated an extensive panel of mouse scTNF mutants... Why not give specific numbers?

- Results: Figure legend S4 - For consistency cytokines should expressed either as μ g or IU.

- Discussion: In the discussion the authors refer to the study by the group of Ruth Ganss, but completely fail to discuss the main relevance of the study to their findings and the mechanistic implications for their own work. Because the work cited aims exactly at the same cytokines (TNF and Interferon) and also achieves vascular targeting without overt toxicity. Ganss claims that TNF normalizes vessels while Interferon is angiostatic. The work by the Ganss group may also explain improved extravasation and the synergy in the CAR-T cell model presented in this study and it also is in line with the more potent anti-angiogenic effects of the Interferon experiments presented. Different, however, from the authors of this study Ganss used primary tumor models of pancreatic cancer (Johansson A et al. *PNAS* 2012).

- Discussion: In relation to the vascular targeting peptides used by the group of Angelo Corti, therapeutic efficacy with 9 μ g TNF in B16 (and RMAS) is very similar to the therapeutic efficacy described in this study and weight loss is only around 10% (their citation 18). Please discuss relative efficiency/toxicity to Acatkine.

- Materials and Methods: Extending on the major critique, to designate the spontaneous mouse melanoma B16 used in this study, a "B16"-line by using the term B16B16 is somewhat misleading, since it originated in 1955 from a B6 mouse and was passaged several hundred times in vivo thereafter. What the genetic makeup of this cell line has to do with the genotype of C57BL/6 mice living today is not clear. Even though the authors used this designation already in their first work analyzing synergy of TNF and IFN more that 30 years ago - explicitly claiming that B16 is "syngeneic" to B6, after these passages and 50 years of strain evolution to claim an identical genetic make up, that is so immunologically compatible that it would not provoke an immune response is hard to uphold. Calling the line B16 (and indicating that it was derived from a B6-mouse in the Materials section) should suffice.

- Materials and Methods: Along the same line, the mice used in this study are not sufficiently described. What is the genetic background of the mice used? For example: Are del-Cre mice referred to, the Tg(CMV-cre)1Cgn mice from the Rajewsky lab (with the CMV-promoter driving the cre recombinase)? If so, than they were probably originally generated in BALB/c - and how often were these mice backcrossed to B6? Please describe the genetic background of all mice used in this study in more detail.

- Materials and Methods: CAR T cell model - Mice were treated when tumors reached the appropriate size. Please specify what is appropriate.

- Throughout the manuscript Abbreviations are excessive: Page 4 line 5, lists a number of abbreviations, which seems confusing. Similarly, the abbreviation AFR is introduced on pages 4, 5 and 10. A clear non-redundant and more restricted use of abbreviations would in my opinion make easier reading, without the need for double-checking the meaning of abbreviations.

1st Revision - authors' response

31 October 2019

Referee #1 (Comments on Novelty/Model System for Author):

The study is based on sophisticated animal models allowing systematic and mechanistic testing of hypotheses in *in vivo* conditions. The approach of reducing the binding affinity of a cytokine towards its receptor and combining it with a targeting moiety to enhance ligand-receptor interaction and thus restore high biological activity deserves more attention from the scientist community.

Ethical concerns: The study contains a substantial number of experiments where 100% lethal dose was determined for various TNF variants. It was not necessary to make the point that the developed mutant is not toxic. This should have been avoided. While I am not an ethicist, it should be brought to the attention of the authors.

Referee #1 (Remarks for Author):

The manuscript by Huyghe and colleagues describes the development and pre-clinical potential of low-affinity, targeted cytokine-based therapeutics against certain solid tumours. The manuscript is very clearly written and the approaches and results are potentially very significant.

I have 3 main comments regarding the work:

1. It would be important to corroborate the claim that the Y86F TNF mutant upon fusion to a targeting antibody regains its adequate biological activity. Assays, more directly probing TNFR1 activation, such as TNFR1 complex formation and characterisation of key molecular components of the complex in parallel with that of induced by WT TNF would confirm that the mutant activate native TNF signalling.

→ This is an interesting and important point. We have already generated substantial data indicating that the targeted TNF Y86F mutant is capable of stimulating key molecular components of the TNF-R1 signaling pathway, including (i) activation of the cell death pathways in MCF7, L929 and B16B16 cells, (ii) NF- κ B-dependent gene transcription in HUVECs (c-FLIP, cIAP-1, A20, CYLD, IL-1b, IL-8) and (iii) p38 and MK2 MAPK phosphorylation in HUVECs. Interestingly, the targeted TNF Y86F mutant, in saturating conditions, is not necessarily doing this to the same extent as WT TNF (e.g. IL-8 induction in HUVECs, as also mentioned in the manuscript). In fact, while treatment with high-dose WT TNF in the B16 model causes destruction of tumor vasculature through induction of EC apoptosis, treatment with high-dose CD13-AFR did not result in this effect. On the other hand, we were able to mimic the effect of WT TNF by combining CD13-AFR with either a PI3K inhibitor (wortmannin) or a Smac-mimetic (Birinapant), thus shifting the balance towards cell death. This suggests that CD13-AFR is capable to induce endothelial cell death *in vivo*, but is hampered to do so by either a negative feedback mechanism (e.g. NF- κ B-dependent anti-apoptotic gene transcription) and/or inefficient complex II formation (e.g. by physical coupling of TNF-R1 to another membrane protein). We certainly plan to investigate this further in greater detail which will require additional extensive datasets, the generation of which will significantly delay publication of this work. Instead, as we believe this is not necessary for the scope of this current article we plan to publish this as a separate manuscript.

2. While it is remarkable that a mutant with a 10,000 fold lower affinity against its target can produce nearly maximal TNFR1 activation, it would be worthwhile to look more into the limitations of such mutants. Most importantly, how would the function of the Y86F mutant

compare with WT TNF in cells expressing low vs high levels of TNFR1? Is there a threshold TNFR1 expression required to induce the biological response?

→ We recognize that this is an interesting issue that needs further investigating, especially since the expression levels of TNF-R1 were shown to be crucial for *in vivo* responses to TNF (e.g. Vanhauwermeiren *et al.*, JCI 2013). This would require careful manipulation and measurement of TNF-R1 expression levels on tumor vasculature, which is rather difficult. Also, even if IFN-g would increase expression levels of TNF-R1 on tumor vasculature, it would be very hard to prove that this effect is responsible for the altered response to CD13-AFR. As for the previous remark, we plan to address this issue in greater detail in a follow-up manuscript.

3. Regarding the *in vivo* assays, while the combination treatments achieved highly synergistic effect, these experiments lack an essential negative control, which is testing the effect of the targeting unit of the construct, namely CD13, CD20. Targeting CD20 has been reported to induce T cell memory and block Tregs (<https://www.ncbi.nlm.nih.gov/pubmed/25231744>) and it is expressed in immune cell populations, plays a role in B cell activation, etc. Similarly, CD13 is an aminopeptidase, with reported functions of inhibiting IL-8-mediated cell death, etc. CD13 is also highly expressed in ovarian cancer cells, thus the effect of the targeting antibodies themselves on tumour eradication (e.g. via ADCC) would need to be evaluated.

→ We agree with the reviewer that we missed a control for the potential effect of the CD13 VHH. Therefore, we have performed an experiment in the B16 model in which we compared the effect of CD13-AFR with a fusion protein consisting of the same CD13 VHH coupled to hIFN α 2 (which has no effect in the B16 model). In the manuscript, Fig.3E was replaced with a new figure including the CD13 VHH control, showing that the CD13 VHH had no effect on tumor growth. Concerning CD13 expression on tumor cells, we did not detect any CD13 expression on B16B16 cells *in vitro* (via flow cytometry and bio-assay with CD13-AFR). Likewise, our mouse CD13 VHH showed no cross-reactivity with human CD13, thereby excluding binding to potential CD13 on human SKOV3 and RL cells. Of note, we could confirm all major *in vivo* findings (except for the combination with CAR T-cells –which was not tested) with an AFR targeted to an oncofetal splice variant of mouse Tenascin-C instead of CD13, thus demonstrating that our findings are not dependent on any CD13-intrinsic function, including its aminopeptidase activity and potential signaling effects.

Referee #2 (Comments on Novelty/Model System for Author):

The authors have used a novel conditional endothelial knockout mouse model, elegant AcTakine constructs, as well as B16B16 tumor cells lacking TNF-R1 or IFN-gamma. All of these unique reagents allowed for a novel and clear research design appropriate to test the main hypothesis.

Referee #2 (Remarks for Author):

Despite of its potent anti-tumor activity, TNF cannot be used therapeutically in view of its extensive systemic toxicity, characterized by hypotension and hepatotoxicity. Apart from the isolated limb perfusion procedure for the treatment of melanoma and soft tissue sarcoma, developed by the teams of Drs Lejeune and Eggermont, the authors have developed an alternative and elegant approach to target the tumor vasculature in melanoma, while reducing systemic toxicity. They make use of perilesional treatment with an Activity-on-Target cytokine (AcTakine) in a mouse B16B16 melanoma model. The AcTakine consists of a single chain mutant Y86F TNF construct, with strongly reduced affinity for TNF receptor 1, fused to an N-terminal VHH against mouse CD13 as a targeting moiety (AFR). This construct prevents unfavorable side and sink effects.

The authors convincingly demonstrate that this AFR AcTakine has an anti-tumor activity towards B16B16 melanoma comparable to wt TNF in conditional endothelial TNF-R1 reactivation mice. In WT mice carrying a TNF-unresponsive B16B16 tumor treated with TNF, the antitumor effect was similar to that observed with a parental B16B16 tumor, thus stressing the importance of TNF-R1 expression in the host cells. Importantly, the recorded systemic toxicity was significantly lower. The authors moreover demonstrate the combined efficacy of tumor vasculature-targeted TNF and IFN-gamma AcTakines to selectively activate or kill tumor endothelial cells. They moreover show that tumor vasculature-targeted AFR therapy synergized with either mouse CD8-AFN immunotherapy,

human CAR T-cell immunotherapy and tumor vasculature-targeted AFN-II therapy. This combination caused a complete eradication of large established tumors in mice without significant toxicity.

In general, this important and high impact study uses state of the art technology, including novel TNF constructs novel conditional TNF-R1 reactivation knockout mice that specifically express TNF-R1 in endothelium, and tumor cells lacking cytokines or their receptors. In general, results are clear and support the main conclusions. I only have some minor remarks with regard to the connection between the *in vivo* and *in vitro* studies.

1. Why were HUVEC cells used as an endothelial cell model in Fig. 2d? Tumor-Associated ECs are mostly microvascular. MVEC have a significantly different phenotype from large vessel endothelial cells with regard to reactivity to LPS and are moreover more sensitive to TNF-induced cell death. This is moreover indicated by the significantly lower IL-8 potential of the CD13AFR in HUVEC cells (Fig 2d), as compared to wt TNF, but a similar induction of ICAM-1 in the tumor microvasculature (Fig. 3c,d). As such, *in vitro* data could be included with commercially available human lung microvascular endothelial cells in order to compare the pro-inflammatory and cytotoxic activities of the constructs with wt TNF in these more relevant endothelial population. Results with mouse MVEC would moreover provide a better connection between the *in vivo* and the *in vitro* results.

→ We completely agree with the reviewer that as a model for tumor vasculature, especially in terms of responses to TNF, HUVECs are not an ideal cell type. However, we used HUVECs, which are CD13-expressing and TNF-responsive cells, to study the effects of a human CD13-targeted TNF mutant *in vitro* (as compared to the non-targeted mutant), not as a model to explain what happens to tumor vasculature. Unfortunately, primary human MVECs are, to our knowledge, CD13-negative, thus preventing their use to address this issue.

2. Although results clearly point towards a crucial role for TNF-R1 in the *in vivo* studies, the role of TNF-R2 is kind of ignored. Yet, despite of not having a death domain, this receptor is important for TNF-induced ICAM-1, VCAM-1 and E-selectin upregulation in endothelium (Chandrasekharan et al., Blood, 2007) and, upon co-activation, can increase cell death induction by TNF-R1, as was also shown in mouse and brain MVEC. The use of hTNF in mice, which does not activate mouse TNF-R2, does not allow this distinction either. Does the Y86F mouse TNF mutant also have reduced binding to TNF-R2?

→ Yes, the TNF Y86F mutant has similarly reduced affinity for TNF-R2, and strongly reduced biological activity in a TNF-R2-dependent CT6 proliferation assay. As neovascular ECs are known to express TNF-R2, we cannot exclude a contribution of this receptor to the effects reported in the manuscript. This is now mentioned in the discussion section of the revised manuscript.

3. Fig S1 demonstrating equal anti-tumor activity of TNF in mice with a wt B16Bl6 or a TNFR1^{-/-} tumor should be included in the main manuscript since these are important data.

→ Fig.S1 is now included in manuscript Fig.1, panel A

4. It is recognized that in Fig. 3b, the sc mutant TNF construct does not induce systemic toxicity *in vivo*, but it also does not induce any IL-6, as compared to controls (the latter of which have rather high basal IL-6 levels in plasma). As such, in this experiment there is no proof that the sc mutant TNF construct has any activity at all.

→ As shown in manuscript Fig.2B, the untargeted (BcII10 VHH) sc mTNF Y86F construct displays biological activity in the L929 cytotoxicity assay at concentrations above 10 ng/ml. To further address this concern, we have performed an additional toxicity experiment (*i.v.* bolus) which also included the 'active' construct CD13-AFR. At therapeutic dose, CD13-AFR did not cause any drop in body temperature, nor a significant increase in plasma IL-6 levels.

5. The coupling of Y86F mTNF and Y87F hTNF to VHH mCD20 or hCD20 induces comparable cytotoxicity as wt TNF in L929 and MCF-7 cells only if these cells express CD20. Yet, in endothelial cells the presence of CD13 does not allow apoptosis induction by the AFR constructs, neither *in vitro* nor *in vivo*, unless when combined with IFN-gamma treatment. Could the expression level of CD13 affect this? Would over-expression of CD13 in EC induce cytotoxicity by the constructs? The authors should discuss this.

→ First, it is important to notice that under normal, rich culture conditions, HUVECs are not sensitive to TNF-induced cell death, even in combination with IFN-g. As such, it is unlikely that over-expression of CD13 in HUVECs would render these cells more sensitive to apoptosis. On the other hand, it is clear that the *in vivo* situation is different: WT TNF can induce EC apoptosis in tumor vasculature, while CD13-AFR appears to be unable to do so. The AcTakine strategy depends on the generation of a high local concentration, thus one would indeed expect that increased expression of the target would render cells more sensitive to the effects of the mutated cytokine, but, as already mentioned above (remark on comment 1 to referee 1), there might be other reasons why the response to CD13-AFR differs from WT TNF. Concerning the combination with CD13-AFN-II/IFN-g, we have no evidence that this alters the expression of CD13, nor is it likely involved since EC apoptosis could already be detected within 2h after a single treatment.

6. How do the authors explain the anti-tumor effect of the AFR in the B16B16 melanoma model (Fig. 3e), despite of the fact that the construct does not induce endothelial apoptosis?

→ The exact mechanism behind the antitumor effect of TNF is complex and not yet fully elucidated. It is clear that EC apoptosis is only part of the mechanism. Other effects may contribute to a greater or lesser extent to the eventual antitumor effect caused by TNF. These may include: (i) haemodynamic effects by increased vessel permeability and/or disseminated intravascular coagulation (DIC), (ii) attraction and infiltration of immune cells (esp. neutrophils) from circulation, and/or (iii) direct anti-angiogenic effects on the developing tumor vasculature.

7. CD13 was recently shown to be expressed not only in tumor vasculature but also in tumor cells, e.g. in patients with soft tissue sarcoma (Kessler et al., *Transl. Oncol.* 2018). As such, could this complicate this approach or confound interpretation of results?

→ Concerning CD13 expression on tumor cells, we did not detect any CD13 expression on B16B16 cells *in vitro* (via flow cytometry and bio-assay with CD13-AFR). Likewise, our mouse CD13 VHH showed no cross-reactivity with human CD13, thereby excluding binding of mouse CD13-AFR to potential human CD13 on SKOV3 and RL cells. To investigate the effect of direct targeting to tumor cells we have performed tumor experiments in which we used B16 cells stably expressing CD20 and treated with CD20-AFR. The effect of CD20-AFR was very similar to CD13-AFR.

Moreover, we could confirm all major *in vivo* findings (except for the combination with CAR T-cells) with an AFR targeted to an oncofetal splice variant of mouse Tenascin-C instead of CD13, thus demonstrating that our findings are not dependent on CD13 aminopeptidase activity and potential signaling effects.

Referee #3 (Comments on Novelty/Model System for Author):

The quality of experiments is good, the novelty medium to high, the impact high. While the mouse models for evaluating toxicity vascular targeting are good, the transplantable tumor models (B16 and the xenogenic tumor) are problematic. The major problem is that the TNF Actakine targets endothelial cells and both transplantable models are very artificial with respect the tumor stroma and building up new vasculature. I think the authors should discuss this limitation critically and also discuss the potential mechanisms for their findings. Currently the manuscript leaves the impression of "marketing" Acatkinines in various combinations. This can in my opinion be overcome by a more detailed and critical discussion of results.

Referee #3 (Remarks for Author):

The study by Huyghe et al. describes an intriguing approach to reduce systemic toxicity, namely by lowering affinity to the cytokine receptor by altering the amino acid sequence and at the same time by targeting the cytokine to a particular cell type or tissue using antibody fragments, they increase the local concentration and thus compensate for lower receptor affinity by increasing avidity of binding in the target tissue.

Expanding on earlier work (using this strategy for type 1 Interferon), in this study they present data for mutated TNF (mTNF) in various combinations of immunotherapy (mTNF alone, mTNF + adoptive CAR T cell therapy, mTNF + IFN γ , mTNF + type 1 Interferon). They use elegant mouse models with tissue specific TNFR1 expression and down modulation of IFN γ R expression to show that progression of transplanted tumor cell lines can be slowed down or tumor cell inocula can even be eradicated without overt toxicity and that TNF action on the CD13 expressing neovasculature of the tumors is important in their model.

While I think the approach of modifying cytokines, as for example the group of Christopher Garcia has shown earlier this year (creating a biased agonist of IFN γ , see Nature 2019), is topical and specifically using a modified TNF in different combinations of immunotherapy is compelling and in principle suitable for publication in EMBO Molecular Medicine, the manuscript in its current form in my opinion is not yet suitable for publication and needs major changes.

Major concern:

My major concern is that while the strategy is interesting and the mouse model for evaluating host effects (toxicity - targeting neovasculature) are appropriate, the transplantable tumor models (B16 and the xenogenic tumor) are problematic. The major problem is that the TNF Actakine approach

targets, a stroma cell population (endothelial cells) and both models are very artificial with respect to the generation of the tumor stroma.

B16 was serially transplanted several hundred times before it became widely available. These types of tumors recruit their stroma in a matter of days, rather than month or years and as stated in Chapter 47 of *Fundamental Immunology* ("Cancer Immunology", 7th edition. Lippincott-Williams & Wilkins, Philadelphia, PA. pages 1200-1234.) "many of these tumors no longer resemble primary mouse or human tumors that virtually always grow at much slower rates". B16 growth is easily affected by so many variables, e.g. another Belgian group has recently shown that just by feeding a high salt diet, growth of this tumor is slowed down (Willebrand et al. 2019 *Frontiers in Immunology*).

The Skov3 xenogenic model suffers from a similar limitation (poor model for microenvironment). The complex heterotypic interactions between tumor and stroma cannot be properly evaluated, because some soluble mediators and surface molecules act across the species barrier while others do not. So again the predictive value is limited.

I think it is clearly beyond the scope of this study (and would be unfair) to ask for data in spontaneous tumor models, but the way the results are presented is not rigorous and critical enough. While transplantable models have their value (as demonstrated by the work of Jim Allison), the authors clearly should discuss more critically the limitations for their work and the preclinical model they used.

→ This is an important point, and to meet this concern we added the statement "the subcutaneous, fast-growing tumor models used in this work might not fully reflect human cancers" to the final paragraph of the discussion section of the manuscript. As for any research, our conclusions only hold true in the models and under the conditions they were verified. It is certainly true that tumor stroma, and possible sensitivity to TNF, strongly varies with tumor growth rate, type and location. We are therefore running or setting more advanced tumor models in our lab, such as the 4T1 orthotopic breast carcinoma model and human patient-derived xenograft models. We would also like to note that other TNF-based therapies that have already advanced to the clinic, such as isolated limb perfusion treatments, or NGR-TNF (the work of Angelo Corti), are based on findings in B16 and other fast-growing tumor models.

Minor points:

1. Results: In vivo Actakines are used daily perilesional for up to 10 days. Did they test immunogenicity of their mutant cytokine fusion proteins? Since feasibility and safety to efficiently eradicate tumors with this treatment is proposed, immunogenicity of the mutated therapeutic proteins needs to be addressed, if not experimentally, it needs to be at least mentioned in the discussion whether it can be ruled out or not.

→ Immunogenicity is a major concern when developing protein drugs, and usually results in severe, acute toxicity shortly after injection of the drug, which is something we have never observed with our constructs. We have selected the TNF Y86F mutant for its low chance of immunogenicity, and VHH's are known to be poorly immunogenic compared to full-sized antibodies. We have, however, not addressed the immunogenicity of our fusion constructs, and have now mentioned this in the discussion section of the revised manuscript.

2. Results: Figure 1, would it not be more accurate to plot temperature graphs as lines of individual mice, instead of substituting 20 degrees for dead mice?

→ It might be more accurate, but, in our opinion, it would be less informative, and it would also require the graphs to be split up per dose. Below (right) is a graph in which temperatures of individual mice were plotted next to the original graph (left). Moreover, one can deduce when mice were found dead or were euthanized from the Kaplan-Meier plots that accompany the temperature graphs.

3. Results: Figure legends, in general it is difficult to follow group size (e.g. 4 to 10 per group is too broad) and the number of biological replications of *in vivo* experiments.

→ Each *in vivo* experiment was repeated at least once (tox experiments) or twice (tumor experiments). In Fig.1D group sizes are indeed broad because these data were pooled from 2 experiments (with group sizes of 4 to 6 mice, depending on availability from breeding) in which different doses were tested. For control reasons, each experiment contained at least one group of the same dose, resulting in some groups of 10 mice in the pooled data. The legend of Fig.1D was adapted to clarify group sizes. For all other *in vivo* experiments, group size is unambiguously mentioned in the figure legend.

4. Results: The FLK (VEGFR2) promoter has been described to be active in endothelial cells. However work by the group of Thomas A. Sake (and others) has also reported activity in other mesodermal cells (e.g. hematopoietic cells and myocytes, see Motoike T. et al. Genesis. 2003). Indeed, when we tested the FLK-cre mice from Georg Breier for recombination in Rosa 26 reporter mice, we found recombination in hematopoietic cells. While Figure 1b suggests that in FLK1-Cre x p55cneo/cneo mice, there is no recombination in spleenocytes, for the novel transgenic mice with FLK1 driven dominant-negative IFN γ R, proof of endothelial specific transgene expression in adult mice is lacking.

→ In Flk1 dominant-negative IFN γ R mice, the dominant-negative effect is dependent on active transcription of the transgene. It is therefore important to distinguish Flk1 promoter activity during development and in the adult mouse. In adult mice, expression of VEGF-R2 is largely restricted to the vascular endothelium, but there are indeed reports showing VEGF-R2 expression in immune cells such as certain types of Tregs and macrophages. We have performed a new IHC analysis, demonstrating that the expression of the dnIFN γ R1 transgene (Myc-tag) coincides largely with PECAM-1/CD31 expression, but not with CD45 expression in B16 tumor, liver and spleen (Fig. EV2D,E). Moreover, we are confident that immune cells are not involved in the synergy of TNF and IFN- γ since it was preserved in athymic nude mice, macrophage-depleted (with clodronate liposomes) mice, and, as reported in the manuscript with CD13-AFR and CD13-AFN-II, also in irradiated NSG mice, which have no functional immune system.

5. Results: Aminopeptidase N (CD13) is expressed on several other tissues besides vascular endothelial cells (such as macrophages, fibroblasts or epithelial cells, see Look A. et al. JCI 1989) and there are also more recent reports of soluble CD13. Therefore, when the authors state that "CD13 enables selective targeting to the tumor vasculature", have they ruled out recognition of other cell types. While they exclude weight loss and overt/apparent toxicity, other on target toxicity or side effects cannot be excluded formally if it is not specifically analyzed. Please discuss.

→ No, we cannot exclude binding of our mouse CD13 VHH to other CD13-expressing cells such as renal and intestinal epithelium and myeloid cells, nor with soluble CD13. It should however be possible to generate and select VHH's that only bind to the neovasculature form of CD13 (in analogy to the NGR peptide, and the hCD13 mAb clone WM15), and we are currently trying to do so, focusing on both mice and man.

6. Results: the p55^{cneo/cneo} model is a very elegant genetic tool and very well used in this study. But it may need an additional cartoon in figure 1b, for better comprehension to the reader that is less familiar with complex compound mutant mouse lines. The term "conditional TNFR1 reactivation knockout" may describe it but seems not self-explanatory.

→ While we agree that these mouse crossings may be complicated for some readers working in other fields, we believe that the principle of crossing conditional knockout mice with Cre mice is very common, and that the sentence "(p55^{cneo/cneo} mice) in which TNF-R1 expression is restored upon Cre-mediated excision of an inhibitory floxed Neo cassette" together with the reference (Victoratos *et al.*, Immunity 2006), in which a scheme can be found, should suffice.

7. Results: Page 5, Acatfactors, ...we evaluated an extensive panel of mouse scTNF mutants... Why not give specific numbers?

→ We have tested 60 hTNF mutants and 44 mTNF mutants, and additional combined mutations in single-chain versions of the cytokines.

8. Results: Figure legend S4 - For consistency cytokines should expressed either as μg or IU.

→ We prefer to keep mIFN γ expressed in IU. 10000IU is around 86ng. The specific affinity can be found in the Methods sections.

9. Discussion: In the discussion the authors refer to the study by the group of Ruth Ganss, but completely fail to discuss the main relevance of the study to their findings and the mechanistic implications for their own work. Because the work cited aims exactly at the same cytokines (TNF and Interferon) and also achieves vascular targeting without overt toxicity. Ganss claims that TNF normalizes vessels while Interferon is angiostatic. The work by the Ganss group may also explain improved extravasation and the synergy in the CAR-T cell model presented in this study and it also is in line with the more potent anti-angiogenic effects of the Interferon experiments presented. Different, however, from the authors of this study Ganss used primary tumor models of pancreatic cancer (Johansson A *et al.* PNAS 2012).

→ Indeed, the work of the group of Ruth Ganss is largely in accordance with our work and may provide a partial mechanistic explanation for our findings. Two important differences with our work, besides the elegant RIP1-Tag5 model for spontaneous pancreatic cancer they have used, are the use of WT cytokines, which allowed signaling on any cell within the tumor, and, an RGR peptide as targeting moiety. The latter is targeting to PDGF-R β which is mainly expressed on pericytes and fibroblasts (which are a significant portion of pancreatic tumors) and is especially enriched in pancreatic cancer. Their claim that TNF induces tumor vessel normalization is interesting, but appears paradoxal to numerous other reports showing that TNF either increases vessel permeability (both *in vitro* as *in vivo*) or causes tumor vessel destruction. We have tested the effect of CD13-AFR on tumor perfusion in the B16 model (using Evans blue) and found that, at 2h after injection, CD13-AFR caused a significant reduction in tumor vessel perfusion, similar to WT TNF.

10. Discussion: In relation to the vascular targeting peptides used by the group of Angelo Corti, therapeutic efficacy with $9\mu\text{g}$ TNF in B16 (and RMAS) is very similar to the therapeutic efficacy described in this study and weight loss is only around 10% (their citation 18). Please discuss relative efficiency/toxicity to Acatkine.

→ The approach used by the group of Angelo Corti is an immunocytokine approach, in which toxicity is reduced by coupling WT TNF to a CD13-targeting NGR peptide, thereby lowering the dose required to reach therapeutic efficacy and reducing toxic side effects. Hence, the relative efficiency/toxicity ratio is increased. In the AcTakine approach, wherein an 'inactivated' form of TNF is used, the efficiency/toxicity ratio is increased by eliminating toxicity in the first place. At therapeutic levels of CD13-AFR, no systemic toxicity (weight loss, body temperature) or systemic activity (IL-6 in plasma) were detectable. Although the results in the B16 and RMA models reported by the group of Angelo Corti are very impressive, the use of NGR-TNF is limited due to a dose-optimum and an inherent risk of side-effects, while this is much less the case for CD13-AFR.

11. Materials and Methods: Extending on the major critique, to designate the spontaneous mouse melanoma B16 used in this study, a "B16"-line by using the term B16B16 is somewhat misleading, since it originated in 1955 from a B6 mouse and was passaged several hundred times *in vivo* thereafter. What the genetic makeup of this cell line has to do with the genotype of C57BL/6 mice living today is not clear. Even though the authors used this designation already in their first work analyzing synergy of TNF and IFN more than 30 years ago - explicitly claiming that B16 is "syngeneic" to B6, after these passages and 50 years of strain evolution to claim an identical genetic make up, that is so immunologically compatible that it would not provoke an immune response is hard to uphold. Calling the line B16 (and indicating that it was derived from a B6-mouse in the Materials section) should suffice.

→ The B16B16 line is a subclone of the B16F10 line that was selected by invasion/migration through mouse bladder 6 times (Poste *et al.*, Cancer Res. 1980).

12. Materials and Methods: Along the same line, the mice used in this study are not sufficiently described. What is the genetic background of the mice used? For example: Are del-Cre mice referred to, the Tg(CMV-cre)1Cgn mice from the Rajewsky lab (with the CMV-promoter driving the cre recombinase)? If so, than they were probably originally generated in BALB/c - and how often were these mice backcrossed to B6? Please describe the genetic background of all mice used in this study in more detail.

→ The Del-Cre mice in our study has the Cre recombinase under control of a rat nestin promoter. These mice were generated in C57BL/6 x CBA background and backcrossed to C57BL/6. (Betz *et al.*, Curr. Biol. 1996).

13. Materials and Methods: CAR T cell model - Mice were treated when tumors reached the appropriate size. Please specify what is appropriate.

→ Treatment was started when tumors reached a size of 4 to 7mm in diameter. This is now mentioned in the Materials and Methods section.

14. Throughout the manuscript Abbreviations are excessive: Page 4 line 5, lists a number of abbreviations, which seems confusing. Similarly, the abbreviation AFR is introduced on pages 4, 5 and 10. A clear non-redundant and more restricted use of abbreviations would in my opinion make easier reading, without the need for double-checking the meaning of abbreviations.

→ Since only two new abbreviations are introduced in the manuscript (AFR and AFN-II) and all other abbreviations are commonly used in biomedical literature, we would like to hold on to the abbreviations used in the manuscript.

2nd Editorial Decision

20 November 2019

Thank you for the submission of your revised manuscript to EMBO Molecular Medicine. We have received the referees' reports, and as you will see they are now supportive of publication of your study pending minor revisions. I am therefore pleased to inform you that we will be able to accept your manuscript once the following points will be addressed:

1) Please address the points raised by referee #1:

Comment #1: please add the information and discuss the concern.

Comment #2: please experimentally address this point.

Comment #3: at this stage, we will not ask you to perform in vivo experiments. If you have data at hand, we would be happy for you to include it. Otherwise, please discuss this referee's concern.

I look forward to reading a new revised version of your manuscript as soon as possible.

***** Reviewer's comments *****

Referee #1 (Comments on Novelty/Model System for Author):

The authors used a range of genetically-modified cell lines and mouse model to allow mechanistic studies. The shortcoming of the studies is more rigorous use of negative controls to test off-target effects of the generated CD13/CD20-VHH-AFN-II and AFR constructs.

Referee #1 (Remarks for Author):

I have two comments on the response to my original queries:

1. Regarding the mechanism of action of the TNF-ANR, the argument that many downstream effect of the low-affinity TNF and WT TNF are shared indicates a similar mechanism of action, nonetheless, the two molecules have significant differences as the authors also elude to, such as the requirement of PI3K or cIAP-inhibition together with CD13-AFR to mimic the effect of WT TNF. These findings should be incorporated in the manuscript and these differences discussed.
2. As the authors highlighted, addressing the required level of TNF-R1 expression for efficacy of the CD13-AFR in vivo is very challenging. Nonetheless, simple in vitro experiments can provide a good insight to this question by correlating TNF-R1 (and for completeness TNF-R2) surface expression of cell lines with the ability of (CD13-HH)-TNF-Y86F to activate TNFR-singalling.
3. Inclusion of negative control (constructs targeting only CD13 and CD20). The additional experiment added in Figure 3E shows that low concentration of CD13-VHH does not have an activity. However, the concentration used was half of that of mCD13-VHH-AFR and only approx one third of the total mCD13-VHH dose used in some combination treatments, thus the effects cannot be compared and it cannot be concluded that CD13-targeting has zero effect. Also, the experiments carried out with Tenascin C targeting are not included in the manuscript.

Referee #2 (Remarks for Author):

The authors have addressed all my concerns in a satisfactory manner. The manuscript was significantly improved.

Referee #3 (Remarks for Author):

The authors have adequately addressed my concerns.

Additional correspondence

21 November 2019

AUTHOR QUERRY:

We have some questions concerning the comments of referee #1:

-I would like to ask you for an exemption for comment #2. This question is far more difficult than the referee makes it appear. It would require very sensitive measurement (e.g. flow cytometry) and manipulation (e.g. via siRNA or overexpression) of TNF receptor expression levels and correlate this with a sensitive readout of TNF receptor activation (e.g. NF- κ B reporter cell line). This is technically very challenging and would require multiple months of work. In the best possible scenario, such experiments would only indicate that certain signaling events require a certain threshold of TNF receptor expression, which is something already suggested in literature (Van

Hauwermeiren et al., JCI 2013), but it would not explain the differential effect of wtTNF and AFR. Also, we believe that this question is not relevant for the main message of the paper.
 -Comment #3 is not justified. The doses given in Fig.3E are in fact as such that exactly the same amount (equimolar) of mCD13 VHH was given in the mCD13-AFR and control group. Moreover, in the combination treatment in the same model (B16), 2.5x less mCD13 VHH is given (daily, more than 4x less when considering the whole treatment period), making it very unlikely that any effect can be attributed to the mCD13 VHH. If possible, we prefer not to include the Tenascin C data.
 -We are willing to include the extra data asked for in comment #1.

We can address all other points within a few days.

EDITOR RESPONSE:

Thank you for your email asking for clarifications. I contacted reviewer #1, who agreed that while point #2 would be interesting to address, it was not essential for publication of the manuscript. Moreover, regarding point #3, this referee realized that the legend indicated equimolar concentration. Therefore, only point #1 would need to be addressed.

2nd Revision - authors' response

29 November 2019

Referee #1 (Comments on Novelty/Model System for Author):

The authors used a range of genetically-modified cell lines and mouse model to allow mechanistic studies. The shortcoming of the studies is more rigorous use of negative controls to test off-target effects of the generated CD13/CD20-VHH-AFN-II and AFR constructs.

Referee #1 (Remarks for Author):

I have two comments on the response to my original queries:

1. Regarding the mechanism of action of the TNF-ANR, the argument that many downstream effect of the low-affinity TNF and WT TNF are shared indicates a similar mechanism of action, nonetheless, the two molecules have significant differences as the authors also elude to, such as the requirement of PI3K or cIAP-inhibition together with CD13-AFR to mimic the effect of WT TNF. These findings should be incorporated in the manuscript and these differences discussed.

→ We have added the data showing the synergistic effect of CD13-AFR and PI3K inhibition (wortmannin) or cIAP inhibition (Birinapant) to the manuscript (Figure 3F,G).

2. As the authors highlighted, addressing the required level of TNF-R1 expression for efficacy of the CD13-AFR in vivo is very challenging. Nonetheless, simple in vitro experiments can provide a good insight to this question by correlating TNF-R1 (and for completeness TNF-R2) surface expression of cell lines with the ability of (CD13-HH)-TNF-Y86F to activate TNFR-singalling.

→ It is indeed to be expected that the expression levels of both the target (CD13) and TNF receptors will influence the cellular response to CD13-AFR. However, tackling this question is quite challenging. It would require very sensitive measurement (e.g. flow cytometry) and manipulation (e.g. via siRNA or overexpression) of TNF receptor expression levels and correlate this with a sensitive readout of TNF receptor activation (e.g. NF-kB reporter cell line). As we have only limited experience with these techniques, it would require multiple months of work. In the best possible scenario, such experiments would only indicate that certain signaling events require a certain threshold of TNF receptor expression, which is something already suggested in literature (Van Hauwermeiren *et al.*, JCI 2013), but it would not explain the differential effect of wtTNF and AFR on endothelial cells. Moreover, we believe that this question is not very relevant for the main message of the paper.

→ After discussion with the editor and referee, it was agreed that while this question would be interesting to address, it was not essential for publication of the manuscript, and therefore no further action was undertaken.

3. Inclusion of negative control (constructs targeting only CD13 and CD20). The additional experiment added in Figure 3E shows that low concentration of CD13-VHH does not have an activity. However, the concentration used was half of that of mCD13-VHH-AFR and only approx one third of the total mCD13-VHH dose used in some combination treatments, thus the effects cannot be compared and it cannot be concluded that CD13-targeting has zero effect. Also, the experiments carried out with Tenascin C targeting are not included in the manuscript.

→ The doses used in the experiment shown in Fig.3E were chosen as such that exactly the same amount (equimolar) of mCD13 VHH was given in the mCD13-AFR and control groups. Moreover, in the CD13-AFR plus CD13-AFN-II combination treatment in the same model (B16Bl6), 2.5 times less mCD13 VHH was given (daily, more than 4 times less when considering the whole treatment period), making it very unlikely that any effect could be attributed to the mCD13 VHH. We prefer not to include the Tenascin C data as we consider using this data in another manuscript.

Referee #2 (Remarks for Author):

The authors have addressed all my concerns in a satisfactory manner. The manuscript was significantly improved.

Referee #3 (Remarks for Author):

The authors have adequately addressed my concerns.

Corresponding Author Name: Jan Tavernier

Manuscript Number: EMM-2019-11223